# Self-MoE: Towards Compositional Large Language Models with Self-Specialized Experts

**Junmo Kang**[*]
Georgia Tech

**Leonid Karlinsky**
MIT-IBM Watson AI Lab

**Hongyin Luo**
MIT

**Zhen Wang**
UCSD

**Jacob Hansen**
MIT

**James Glass**
MIT

**David Cox**
MIT-IBM Watson AI Lab

**Rameswar Panda**
MIT-IBM Watson AI Lab

**Rogerio Feris**
MIT-IBM Watson AI Lab

**Alan Ritter**
Georgia Tech

## Abstract

We present Self-MoE, an approach that transforms a monolithic LLM into a compositional, modular system of self-specialized experts, named MiXSE (MiXture of Self-specialized Experts). Our approach leverages self-specialization, which constructs expert modules using self-generated synthetic data, each equipping a shared base LLM with distinct domain-specific capabilities, activated via self-optimized routing. This allows for dynamic and capability-specific handling of various target tasks, enhancing overall capabilities, without extensive human-labeled data and added parameters. Our empirical results reveal that specializing LLMs may exhibit potential trade-offs in performances on non-specialized tasks. On the other hand, our Self-MoE demonstrates substantial improvements (6.5%p on average) over the base LLM across diverse benchmarks such as knowledge, reasoning, math, and coding. It also consistently outperforms other methods, including instance merging and weight merging, while offering better flexibility and interpretability by design with semantic experts and routing. Our findings highlight the critical role of modularity, the applicability of Self-MoE to multiple base LLMs, and the potential of self-improvement in achieving efficient, scalable, and adaptable systems.

## 1 Introduction

The remarkable success of Large Language Models (LLMs) has been largely attributed to their generalist nature, allowing them to perform a wide variety of tasks (Brown et al., 2020; Touvron et al., 2023; Jiang et al., 2023; Team et al., 2024). Predominantly designed as monolithic architectures, these models rely extensively on large-scale data to embed generalized language capabilities across vast parameter spaces. While effective, this monolithic architecture, as illustrated in Figure 1, inherently suffers from significant drawbacks such as inefficiency in scaling (Zhang et al., 2024; Wan et al., 2024), susceptibility to forgetting previously learned information when adapted to specialized tasks (Kotha et al., 2024; Huang et al., 2024), and a lack of transparency which leads to the black-box nature (Zhao et al., 2023).

Meanwhile, the increasing demand to handle domain-specific or expert-level tasks has highlighted the need for specialization of LLMs (Cheng et al., 2024; Ling et al., 2023; Feng et al., 2024). However, effective tuning often relies on high-quality, human-annotated data, which is costly and challenging to scale (Kang et al., 2023), especially in specialized domains where expertise is scarce and valuable (Wu et al., 2023). Self-specialization (Kang et al., 2024) offers a promising alternative, aligning models with self-generated synthetic data. While this technique has proven effective in cross-task generalization within a target expert domain, we posit that it may compromise performance in areas outside the target domain.

---

[*]Correspondence to `junmo.kang@gatech.edu`

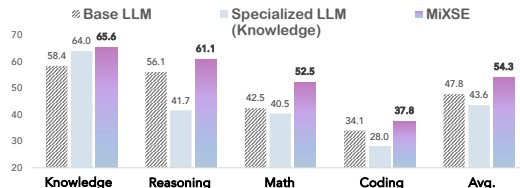

Figure 1: Concept of Self-MoE, illustrating the transformation from a monolithic LLM to a compositional system, MiXSE, without extensive resources and addition of significant parameters. MiXSE distinguishes itself from traditional MoEs and other models in post-training, lightweight semantic experts, and/or self-generated synthetic data. The results showcase MiXSE's improved capabilities over the base LLM (e.g., Gemma-7B) across all domains, unlike the knowledge-specialized LLM that compromises other capabilities.

In this paper, we explore the following question: *How can we build compositional LLMs that enjoy versatile expertise, while using minimal resources?* We introduce Self-MoE (Figure 1), an approach that transforms a monolithic model into a compositional (Zaharia et al., 2024) system, called MiXSE (MiXture of Self-specialized Experts). This approach differs from prior MoE work using LoRA (Hu et al., 2022), which either relies on human-labeled data (Wu et al., 2024) or assumes the existence of trained modules (Huang et al., 2023; Muqeeth et al., 2024). Instead, our Self-MoE constructs individual lightweight expert modules from scratch using synthetic data, inspired by the concept of self-specialization. Each module is integrated with a shared base LLM, and the entire system is enhanced by a self-optimized routing mechanism. In contrast to monolithic models, which often suffer from forgetting issues when adapted or merged under fixed, static parameters, our modular design preserves the integrity and semantics of each expert. This allows for dynamic, precise handling of various target domain tasks, boosting the model's overall capability, adaptability, and interpretability.

Through extensive empirical studies conducted across a variety of popular domains, including knowledge, reasoning, math, and coding, we find that specialization often comes with trade-offs, typically degrading performance in non-targeted domains. However, our Self-MoE demonstrates substantial overall improvements over a base LLM across all target domains without compromising performance on other tasks. Notably, the compositional nature of our MiXSE appears to exploit synergies among experts, even outperforming all individual specialized experts.

Moreover, MiXSE clearly surpasses other strong baselines such as instance merging and weight merging, under similar settings, while offering better flexibility and interpretability. Detailed analyses highlight the critical role of the routing mechanism and the contribution of semantic experts in achieving these results. Our interpretable visualizations of routing distributions further elucidate how tasks are dynamically allocated to the most relevant experts. Lastly, we further validate that there are no issues related to forgetting unlike monolithic baselines, and that our approach can be applied to various model families and sizes. In summary, our key contributions are as follows:

- We highlight the inherent limitations of monolithic model specialization, where focusing on a specific capability often comes at the cost of degrading performance in other domains.

- We propose Self-MoE, which allows a base, monolithic LLM to upgrade into a modular system of lightweight, self-specialized experts, without requiring extensive human supervision, compute resources, or overhead in active parameters.

- We provide comprehensive experiments and analyses across a range of benchmarks, where Self-MoE demonstrates consistent improvements with an average of 6.5%p across domains over a base LLM, outperforming various baselines. Our ablation studies validate the impact of modularity, routing strategies, and the use of self-generated synthetic data. Moreover, our analyses explore routing distributions, forgetting issues, and the applicability to five different base LLMs.

## 2 PROBLEM STATEMENT

The primary focus of this work is on self-improving LLMs' target capabilities on the fly, specifically under settings constrained by minimal resources and without the addition of significant parameters.

**Self-Specialization**

**MiXSE (MiXture of Self-Specialized Experts)**

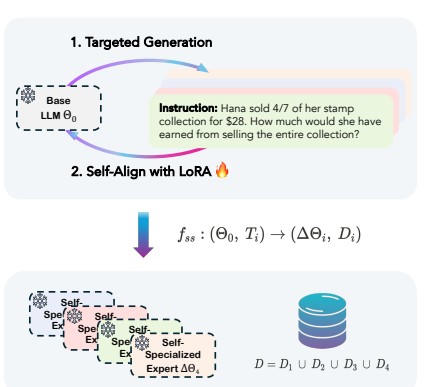
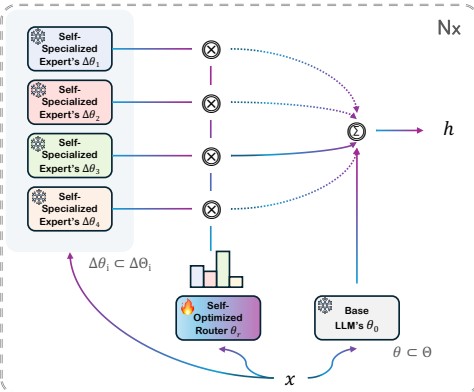

Figure 2: Overview of the **Self-MoE** approach to building a compound system of specialized experts and a router in a self-improving manner. In the Self-Specialization phase (left side), the base LLM is aligned with self-generated synthetic data for each target specialization, producing lightweight expert modules. The right side shows MiXSE where each self-specialized expert is dynamically engaged based on the decisions of the self-optimized router.

Traditional LLMs, which are generally monolithic, require expensive human-labeled data to be better specialized, thereby limiting their adaptability and scalability when resources are constrained. We hypothesize that a modular, compositional model utilizing self-generated synthetic data for self-improvement can dramatically improve specific target capability, adaptability, and interpretability while reducing dependency on expensive human-annotated datasets.

Specifically, given a base LLM $\Theta_0$ and a minimal set of seed data (e.g., 100) for each of the target capabilities $\{T_i\}_{i=1}^n$ (e.g., knowledge, math), our goal is to transform $\Theta_0$ into an enhanced compositional model $\Theta_{comp}$ where $n$ target expert modules $\{\Delta\Theta_i\}_{i=1}^n$ are effectively integrated. Formally, the Self-MoE transformation function is defined as:

$$f_{trans} : (\Theta_0, \{T_i\}_{i=1}^n) \rightarrow \Theta_{comp} = \Theta_0 \cup \{\Delta\Theta_i\}_{i=1}^n$$

Here, under our problem setting, the number of parameters of $\Theta_0$ and $\Theta_{comp}$ should not be significantly different, necessitating that the expert modules $\Delta\Theta_i$ be lightweight (i.e., LoRA (Hu et al., 2022)). The available seed data are limited but can be reasonably collected (e.g., 100). Importantly, we do not assume the availability of larger/teacher models at one's hand; instead, we aim to develop a method that enables self-improvement and is designed to be universally applicable.

## 3    METHOD: SELF-MOE

In this section, we describe Self-MoE, our proposed framework designed to build a compositional model in which specialized expert modules and a routing component are learned in a self-training manner to cooperate effectively. At a high level, Self-MoE decomposes the monolithic structure of a base LLM into a dynamic mixture of self-specialized units, each equipped with distinct target capabilities. This section outlines the overall pipeline and architecture of Self-MoE, illustrated in Figure 2, which details both the self-specialization of individual target expert modules and their integration to form a compositional system, MiXSE (MiXture of Self-specialized Experts).

### 3.1    BUILDING EXPERT MODULES THROUGH SELF-SPECIALIZATION

The first step of Self-MoE is creating specialized modules $\{\Delta\Theta_i\}_{i=1}^n$ for each target expertise, while adhering to the desiderata discussed in Section 2. That is, the modules should be lightweight and self-improving. We employ self-specialization (Kang et al., 2024) where a base LLM is aligned with self-generated data for target specialization, resulting in lightweight LoRA (Hu et al., 2022) experts.

**Targeted Generation.**    Self-specialization involves generating synthetic instruction-response data $D_i = \{(inst_i^{(1)}, resp_i^{(1)}), (inst_i^{(2)}, resp_i^{(2)}), ...\}$ tailored to each target domain $T_i$. We ensure the data

is both diverse and highly relevant to the specialized tasks/domains each module will address. The generation includes the following steps:

**(1) Seed Construction**: First, given a target $T_i$ identified, we prepare a small number of seed examples (e.g., 100) that capture essential characteristics and scenarios relevant to each target domain $T_i$. While we exploit existing datasets for the purpose of demonstration, we posit manual annotation for such a small number should be reasonable in real-world applications. These seeds serve as the foundational dataset from which synthetic variations are generated.

**(2) Instruction Brainstorming**: Once the seed examples are established, the next step is to diversify the range of instructions (and corresponding input contexts) through a brainstorming process. Specifically, we prompt[1] a base LLM $\Theta_0$ to create new instructions following sequences of seed instructions given in-context.

**(3) Response Generation**: The final step involves generating corresponding responses for the newly created instructions. We use seed instruction-response pairs as in-context demonstrations to extract latent relevant knowledge from $\Theta_0$.

**Self-Align with LoRA**    With each specialized synthetic data $D_i$ in place, we now proceed with the self-alignment of $\Theta_0$ to induce specialization, separately producing lightweight expert components $\Delta\Theta_i$. Note that $D_i$ are self-generated by $\Theta_0$ and used to specialize the same $\Theta_0$ using an adapter module $\Delta\Theta_i$, resulting in an specialized model $\Theta_{spec} = \Theta_0 + \Delta\Theta_i$. Specifically, we utilize Low-Rank Adaptation (LoRA) (Hu et al., 2022), which integrates additional trainable parameters that are specific to each domain $T_i$ while keeping $\Theta_0$ intact. Within the corresponding $\Theta$, we define $\theta$ as the weights at a certain layer where LoRA is attached. Let $\theta_{spec} \in \mathbb{R}^{d \times k}$ be updated weights at a specific LoRA layer which can be decomposed as:

$$\theta_{spec} = \theta_0 + \Delta\theta_i$$
$$= \theta_0 + \theta_{B_i}\theta_{A_i}$$

where $\theta_{B_i} \in \mathbb{R}^{d \times rank}$ and $\theta_{A_i} \in \mathbb{R}^{rank \times k}$, with $rank \ll \min(d, k)$. The forward pass becomes:

$$h = \theta_{spec}x = \theta_0 x + \theta_{B_i}\theta_{A_i}x$$

This applies to all LoRA layers, and only $\Delta\Theta_i = \{\Delta\theta_i^{(1)}, \Delta\theta_i^{(2)}, ...\}$ is updated during training using $D_i$. As a whole, this process of self-specialization can be defined as producing an expert module $\Delta\Theta_i$ for the $i$-th target along with the corresponding synthetic data $D_i$ (Left in Figure 2):

$$f_{ss} : (\Theta_0, T_i) \rightarrow (\Delta\Theta_i, D_i)$$

We iterate this process for each target domain, focusing on knowledge, reasoning, math, and coding.

## 3.2 Mixture of Self-Specialized Experts

After each expert module is individually specialized through the self-specialization process, they are integrated into a compound system $\Theta_{comp}$, MiXSE (MiXture of Self-specialized Experts). MiXSE is designed to leverage the distinct capabilities of each module, orchestrating their cooperation to handle diverse tasks dynamically and efficiently. To achieve this benefit, a router module $\theta_r$ is also incorporated, which analyzes each input token to dynamically route to the most appropriate expert module based on the task at hand.

Specifically, within each layer, the output $h$ for each input $x$ is calculated by combining the contributions of the selected expert modules $\Delta\theta_i$, weighted by their relevance determined by the router:

$$h = \theta_0 x + \sum_{i=1}^{n} \alpha_i \Delta\theta_i x$$
$$= \theta_0 x + \sum_{i=1}^{n} \alpha_i \Delta\theta_{B_i}\theta_{A_i}x$$

where $\alpha$ represents a set of weights computed by the router (i.e., a linear layer) $\theta_r \in \mathbb{R}^{n \times k}$.

$$\alpha = \text{top-k}(\text{softmax}(\theta_r x))$$

---

[1]The prompts can be found in Table 11-14 in Appendix.

Note that we only take top-k probabilities and mask out the others to efficiently reduce computation. In essence, this also allows the pre-trained base weights $\theta_0$ to be sufficiently able to contribute, mitigating potential issues of over-specialization such as forgetting or diminished generalizability. The router $\theta_r$ is a linear layer, shared across all LoRA layers, and is trained using the aggregated self-generated data $D = \{D_i\}_{i=1}^{n}$ to learn how to optimally select modules for a given task:

$$L(\theta_r) = -\mathbb{E}_{(inst,\ resp) \sim D}[logP_{\Theta_0}(resp \mid inst; \theta_r, \{\Delta\Theta_i\}_{i=1}^{n})]$$

It can be noted that the router is not provided with explicit supervision about which expert should be selected for each token, as there is no fixed label for each token indicating the correct single expert. Instead, it learns the optimal expert selection indirectly through training on self-generated instructions and responses. The supervision comes from the responses, where the routing decisions are determined dynamically based on the token-level features to allow the model to produce better responses. Importantly, the router is optimized separately after the expert modules are trained and frozen, ensuring the explicit semantic distinction of the expert modules is preserved.

## 4 EXPERIMENTS AND RESULTS

**Datasets.** We evaluate Self-MoE across diverse domains categorized into knowledge, reasoning, math, and coding: MMLU (0- & 5-shot) (Hendrycks et al., 2021a), BBH (3-shot) (Suzgun et al., 2022), GSM8K (8-shot) (Cobbe et al., 2021), and HumanEval (0-shot) (Chen et al., 2021), respectively. For MMLU, we primarily employ the 0-shot setting unless otherwise specified, based on established observations (Dettmers et al., 2023; Lin et al., 2024) that tuning yields only marginal effects in the 5-shot setting for this task. To test generalization (Section 4.4), we additionally evaluate on MATH (4-shot) (Hendrycks et al., 2021b), MBPP (3-shot) (Austin et al., 2021), NaturalQuestions (5-shot) (Kwiatkowski et al., 2019), TriviaQA (5-shot) (Joshi et al., 2017), Hellaswag (0-shot) (Zellers et al., 2019), PIQA (0-shot) (Bisk et al., 2020), and TruthfulQA (0-shot) (Lin et al., 2022).

**Baselines.** To assess the effectiveness of Self-MoE, we compare performance against several baselines that are similarly trained using LoRA and that use the same number of active parameters during inference for fair comparisons:

- Four Self-Specialized Models (Kang et al., 2024): Trained on self-generated synthetic data for individual domains: knowledge, reasoning, math, and coding.
- Instance Merging (Multi-Task Tuning) (Chung et al., 2024): Leverages the aggregated synthetic data generated by self-specialization to train a model capable of handling multiple tasks.
- TIES (Yadav et al., 2023), DARE (Yu et al., 2024): Advanced weight merging methods integrating multiple expert strengths into a unified model.

Note that Self-MoE does not require the base models to be implemented using specific architectures. Rather, Self-MoE builds upon purely any base LLMs using LoRA-based fine-tuning like other baselines, which ensures fair and consistent comparisons. We also contextualize these results with computationally intensive methods reported in the literature, despite indirect comparisons: BTM (Li et al., 2022), Sparse Upcycling (Komatsuzaki et al., 2023), BTX (Sukhbaatar et al., 2024), GLAN (Li et al., 2024a), Orca (Mitra et al., 2023), and Merlinite (Sudalairaj et al., 2024) in Appendix D.1.

**Implementation Details.** We adopt Gemma-7B (Team et al., 2024) as a base LLM for our main experiments, and additionally apply Self-MoE to various models, such as LLaMA-2 7B & 13B (Touvron et al., 2023), Mistral 7B (Jiang et al., 2023), and LLaMA-3 8B (AI@Meta, 2024) in Section 4.5. We use 100 seeds to generate 5K synthetic data for each domain, resulting in 20K data. Each LoRA module contributes less than 0.3% to the parameters of the base model, and the router's parameters are negligible, resulting in the added parameters of MiXSE amounting to only about 1%.

### 4.1 MAIN RESULTS

In Table 1, we showcase comparative benchmark results of various approaches across four specialized domains: knowledge, reasoning, math, and coding. All baselines use self-generated synthetic data based on the same Base LLM, Gemma-7B, and LoRA for tuning to ensure fair comparisons.

Table 1: Main results. All models are built upon the same base LLM, Gemma-7B, taking self-improving approaches and having the same active parameters during inference. Corresponding aligned performances of self-specialization are underscored. Each column's best performance is highlighted in bold, while the gains achieved by our MiXSE over the base LLM are indicated.

| Method | Active Params | Knowledge (MMLU) | Reasoning (BBH) | Math (GSM8K) | Coding (HumanEval) | Avg. |
|---|---|---|---|---|---|---|
| Base LLM | 7B | 58.4 | 56.1 | 42.5 | 34.1 | 47.8 |
| *Specialized LLM for Each Capabiility* | | | | | | |
| Knowledge Self-Spec. | 7B + 0.3% | 64.0 | 41.7 | 40.5 | 28.0 | 43.6 |
| Reasoning Self-Spec. | 7B + 0.3% | 60.1 | 60.2 | 41.0 | 28.7 | 47.5 |
| Math Self-Spec. | 7B + 0.3% | 59.3 | 58.9 | 50.0 | 36.0 | 51.1 |
| Coding Self-Spec. | 7B + 0.3% | 57.2 | 57.2 | 46.0 | 37.2 | 49.4 |
| *Merging Methods* | | | | | | |
| Instance Merging | 7B + 0.3% | 62.6 | 57.6 | **53.5** | 36.0 | 52.4 |
| TIES Merging | 7B + 0.3% | 63.7 | 56.3 | 38.5 | 32.9 | 47.9 |
| DARE Merging | 7B + 0.3% | 37.7 | 59.6 | 45.0 | 34.8 | 44.3 |
| MiXSE (Ours) | 7B + 0.3% | **65.6** ↑7.2 | **61.1** ↑5.0 | 52.5 ↑10.0 | **37.8** ↑3.7 | **54.3** ↑6.5 |

First, we confirm self-specialization markedly enhances target-specific expertise, compared to the base LLM. For instance, we can see substantial gains from corresponding specialized models (e.g., Knowledge Self-Spec. in the knowledge domain): 58.4 to 64.0 in knowledge, 56.1 to 60.2 in reasoning, and so on. However, this focused improvement sometimes comes at the cost of reduced performance in non-targeted areas, as evidenced by the drop in scores for the Knowledge Self-Spec. model in reasoning, math, and coding. This trade-off highlights the inherent limitation of over-specialization. In contrast, our MiXSE, demonstrates consistent improvements across all domains, due to its modular, compositional architecture that makes use of dynamic routing to leverage optimal experts. Surprisingly, it even outperforms all corresponding specialized models, indicating that it effectively synergizes the strengths of each specialization.

In comparison with other static merging methods like Instance Merging, TIES, and DARE, MiXSE stands out for its superior adaptability. While they attempt to combine the strengths of different specialization areas into a single model, they lack the dynamic flexibility that MiXSE offers. Notably, simple instance merging (i.e., multi-task tuning), though effective in enhancing the base LLM across domains, still falls short of achieving the superior average performance of 54.3 seen with MiXSE. This validates the advantages of dynamic expert integration in a compositional system.

## 4.2 ABLATION STUDY

Now that we have verified the effectiveness of MiXSE as a whole, we evaluate the impact of different configurations and components of the system, presented in Table 2. The configurations vary in terms of routing strategies and integration of experts, offering insights into the contributions of each element to the system's overall effectiveness.

We start by examining the Top-k routing strategy, which plays a crucial role in our model. Our findings show that both the Top-1 and Top-2 expert configurations deliver the best performance. This suggests that identifying and leveraging the most relevant expert for a given task is typically sufficient and most effective. On a side note, the similar performances of the different configurations may highlight the robustness of our method. Given the similar performances, we prefer the Top-1 expert setup for better efficiency.

Interestingly, the results also indicate a drop in performance when using All Experts. This can be attributed to that involving all experts regardless of their relevance can introduce noise and dilute the specific contributions of the most pertinent experts. Additionally, involving more experts than necessary can increase computational overhead.

We observe that the performance significantly decreases with random routing (i.e., w/o Self-Optimized Router), highlighting the router's role in dynamically tailoring the selection of experts according to the specific requirements of each task. The router's ability to discern and activate the

Table 2: Analysis and ablation of the router in our MiXSE. Configurations vary to investigate the optimal number of experts used, to verify the possibility of self-learning for the router, and to see the importance of semantic distinctions among experts within the compositional system.

| Configuration | Knowledge (MMLU) | Reasoning (BBH) | Math (GSM8K) | Coding (HumanEval) | Avg. |
|---|---|---|---|---|---|
| Base LLM | 58.4 | 56.1 | 42.5 | 34.1 | 47.8 |
| *Top-k Routing* | | | | | |
| w/ Top-1 Expert | **65.6** | **61.1** | 52.5 | 37.8 | **54.3** |
| w/ Top-2 Experts | 65.5 | 60.9 | 52.5 | **38.4** | **54.3** |
| w/ All Experts | 65.4 | 58.9 | **54.0** | 33.5 | 53.0 |
| *Routing Strategy* | | | | | |
| w/o Self-Optimized Router | 59.9 | 58.5 | 48.0 | 36.6 | 50.8 |
| w/o Shared Router | 59.5 | 59.1 | 50.5 | 32.9 | 50.5 |
| *Experts & Router Joint Training* | | | | | |
| w/o Semantic Experts (Top-1) | 64.5 | 58.1 | 46.0 | 33.5 | 50.5 |
| w/o Semantic Experts (Top-2) | 64.2 | 53.3 | 48.5 | 36.5 | 50.6 |

most suitable experts based on the context is critical for optimizing performance. Notably, this ability is learned by relying on a very small amount of seed data. When employing layer-specific routers instead of the shared router, we found that the performance substantially drops, despite having about 200x more parameters, justifying our choice. This might be attributed to the fact that the layer-specific ones may introduce conflicting routing decisions, possibly requiring more data or hyperparameter tuning to become effective.

Another interesting finding comes from the configuration where experts and the router are jointly trained, which means that the semantic distinctions among experts may be diluted. This setup (w/ either Top-1 or Top-2) substantially decreases performance relative to scenarios where the router and experts are optimized independently. This decline validates that semantic experts play a crucial role in enhancing the system's capability to handle tasks requiring specific expertise, while offering better interpretability (Section 4.3).

## 4.3 ROUTING ANALYSIS

Understanding how MiXSE allocates tasks to its various experts is crucial for gauging its interpretability. By analyzing the routing distributions across four distinct domains, we aim to see whether the system matches queries to the most suitable experts. Figure 3 presents the routing distributions used to solve each benchmark, where the weights are averaged across tokens and layers within individual tasks.

We first observe that the MiXSE's router effectively selects the correct expert for each corresponding target. This is evident from the impressive alignment between tasks and the experts chosen by the router; for example, the knowledge expert predominantly handles knowledge tasks, while the coding expert

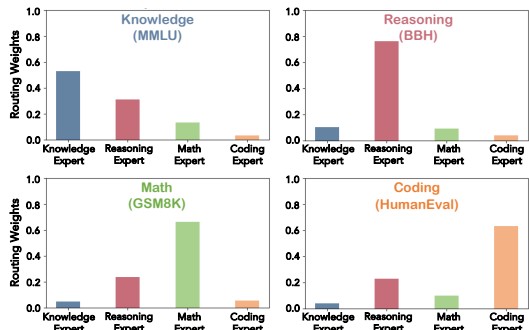

Figure 3: Routing analysis that shows routing distributions over four domains for each benchmark, averaging the weights across tokens within individual tasks.

is routed coding tasks. This highlights the router's ability to learn and apply this routing automatically and consistently, making the system's decisions interpretable and trustworthy.

Beyond the direct matching of tasks to domain-specific experts, the router also demonstrates its ability to exploit synergies between different areas of expertise. For instance, the reasoning expert is frequently involved in tasks across the knowledge, math, and coding, reflecting the system's compositional use of expertise. This explains the reason for MiXSE's superior performances across all domains even beyond all specialized modules in Table 1.

Table 3: Investigation on generalization and a forgetting issue of Self-MoE. Non-Target (In-Expertise) indicates where MiXSE does not directly specialize using seed data directly while relevant to targets. Non-Target (Out-of-Expertise) refers to irrelevant cases.

| Category | Benchmark | Base LLM | Instance Merging | MiXSE |
|---|---|---|---|---|
| *Target* | | | | |
| Academic Knowledge | MMLU | 58.4 | 62.6 | 65.6 |
| Reasoning | BBH | 56.1 | 57.6 | 61.1 |
| Math | GSM8K | 42.5 | 53.5 | 52.5 |
| Coding | HumanEval | 34.1 | 36.0 | 37.8 |
| Target Average | | 47.8 | 52.4 | 54.3 |
| *Non-Target (In-Expertise)* | | | | |
| Math | MATH | 20.7 | 15.3 | 21.4 |
| Coding | MBPP | 37.8 | 37.6 | 39.6 |
| *Non-Target (Out-of-Expertise)* | | | | |
| World Knowledge | Natural Questions | 24.2 | 22.3 | 24.5 |
| | TriviaQA | 63.9 | 58.6 | 62.5 |
| Commonsense | Hellaswag | 80.6 | 78.0 | 80.7 |
| | PIQA | 81.1 | 80.1 | 81.2 |
| Safety | TruthfulQA | 44.7 | 42.2 | 44.3 |
| Non-Target Average | | 50.4 | 47.7 | 50.6 |

## 4.4 GENERALIZABILITY TEST

While Self-MoE has shown clear benefits in target benchmarks such as MMLU, BBH, GSM8K, and HumanEval, one may be curious about its generalizability to non-targets, or concerned with the potential issues of specialization such as forgetting. In Table 3, we conduct an investigation using non-targeted benchmarks that were not utilized in building MiXSE.

On MATH and MBPP benchmarks, which can be considered highly relevant to target benchmarks, GSM8K and HumanEval, we find our Self-MoE can still improve over the base LLM even though they were not directly targeted in our training regime, indicating generalizability.

Concerning the potential side effect of forgetting, we extend our testing to include domains such as world knowledge, common sense, and safety, which are rarely associated with the targets directly. Our experiments show that overall, there are rarely meaningful performance drops when applying our Self-MoE. Only a minor drop is observed with MiXSE in TriviaQA, but this is substantially less than in the case of instance merging. This suggests our approach almost maintains existing knowledge for non-targets while significantly boosting target performances, unlike monolithic baselines.

## 4.5 APPLICABILITY TO OTHER BASE LLMS

Following the successful demonstration of our Self-MoE approach based on Gemma-7B, we now present Figure 4 where we apply Self-MoE to other base LLMs beyond Gemma-7B. We use diverse model variants including LLaMA-2 7B & 13B, Mistral 7B, and LLaMA-3 8B. Our findings suggest that our approach improves all models on average regardless of the model family, size, and level of base performance, outperforming the strong instance merging baseline.

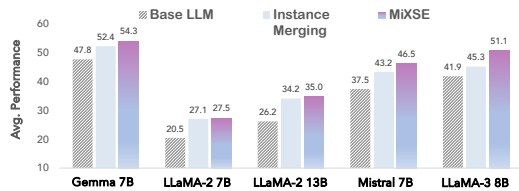

Figure 4: Results of Self-MoE w/ other LLMs.

## 4.6 IMPACT OF THE NUMBER OF SYNTHETIC DATA

Figure 5 illustrates the impact of scaling self-generated synthetic data for Self-MoE. As the data scales from 0 to 20K, our MiXSE model demonstrates substantial and consistent improvements over the base one in average performance across domains, suggesting the scalable potential of Self-MoE. Instance Merging, serving as a strong baseline, also benefits from increased data, but the gains progress at a slower rate, as evidenced by linear trendlines. This reflects the inefficiency of the static merging scheme, which, being monolithic, suffers from trade-offs in knowledge gains and forgetting.

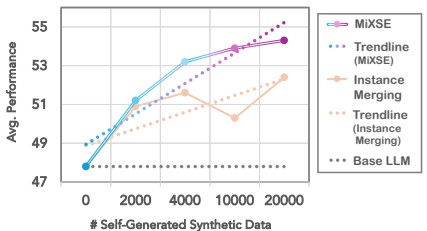

Figure 5: Analysis with the varied sizes of self-generated synthetic data.

## 4.7 Scaling the Number of Experts

In Table 4, we present the results of MiXSE composed of varying numbers of experts, with experts added progressively one at a time in the order of knowledge, reasoning, math, and coding. The results indicate that starting with the knowledge expert, which initially exhibits a performance trade-off, subsequent additions of reasoning, math, and coding experts consistently enhance overall performance.

Table 4: Scaling the number of experts. K: Knowledge expert. R: Reasoning expert. M: Math expert. C: Coding expert.

| # Experts | Knowledge (MMLU) | Reasoning (BBH) | Math (GSM8K) | Coding (HumanEval) | Avg. |
|---|---|---|---|---|---|
| 0 (Base LLM) | 58.4 | 56.1 | 42.5 | 34.1 | 47.8 |
| 1 (K) | 64.0 | 41.7 | 40.5 | 28.0 | 43.6 |
| 2 (K+R) | 65.8 | 58.0 | 43.0 | 32.3 | 49.8 |
| 3 (K+R+M) | 62.7 | 61.5 | 54.5 | 32.9 | 52.9 |
| 4 (K+R+M+C) | 65.6 | 61.1 | 52.5 | 37.8 | 54.3 |

This highlights the compositional MiXSE's advantage of adaptability and modularity.

## 4.8 Analyses on Self-Generated Synthetic Data

We conduct analyses of the self-synthesized datasets in Table 5. For diversity measurement, we first analyze the linguistic diversity using Type-to-Token Ratio (TTR), and the semantic similarity of the pairwise instructions' embeddings using SBERT (Reimers & Gurevych, 2019). Synthetic data demonstrates comparable linguistic diversity to human-labeled data, with a slightly higher TTR for BBH, suggesting that the syn-

Table 5: Analyses of self-generated synthetic data in terms of diversity, complexity, and naturalness.

| Metric | Knowledge (MMLU) | Reasoning (BBH) | Math (GSM8K) | Coding (HumanEval) | Avg. |
|---|---|---|---|---|---|
| *Type-to-Token Ratio (TTR) (↑)* | | | | | |
| Human-Labeled Data | 0.2671 | 0.1672 | 0.1683 | 0.1121 | 0.1787 |
| Synthetic Data | 0.2639 | 0.1889 | 0.1484 | 0.0961 | 0.1743 |
| *Semantic Similarity (↓)* | | | | | |
| Human-Labeled Data | 0.2625 | 0.1757 | 0.4125 | 0.4608 | 0.3279 |
| Synthetic Data | 0.3129 | 0.1948 | 0.3360 | 0.4791 | 0.3307 |
| *Classification Accuracy (↓)* | | | | | |
| LLM as-a-judge (GPT-4o) | 55.0 | 68.0 | 60.0 | 50.0 | 58.3 |
| *Model Performance using Different Data (↑)* | | | | | |
| w/ Human-labeled data (Seed) | 57.4 | 57.0 | 45.0 | 34.1 | 48.4 |
| w/ Synthetic data (1x) | 57.7 | 55.9 | 45.5 | 32.9 | 48.0 |
| w/ More Synthetic data (5x) | 61.3 | 58.4 | 48.4 | 36.6 | 51.2 |
| w/ More Synthetic data (50x) | 65.6 | 61.1 | 52.5 | 37.8 | 54.3 |

thetic data includes richer lexical variation, especially in reasoning tasks. For semantic similarity, synthetic data achieves generally low similarity within each dataset, similar to human-labeled data (0.3307 vs. 0.3279) on average. This suggests a high semantic diversity overall, reflecting the natural diversity found in human-labeled data.

Next, we leverage a strong instruction-following model, GPT-4o, as a judge to classify which instruction was synthetic. Given 100 pairs of human-labeled and synthetic instructions, the classification accuracy ranged from 50% (random guessing) to 68%, with the lowest accuracy for HumanEval and MMLU, suggesting that synthetic data closely mimics human complexity and naturalness in these domains. Conversely, the higher accuracy for BBH and GSM8K indicates that synthetic data in these domains has room to improve.

Finally, we compare the performance of Self-MoE fine-tuned with synthetic data against human-labeled seed data. We observe that with the same quantity (400) as the seed, synthetic data achieves performance similar to human-labeled data on average. When scaling up the size (5x and 50x), synthetic data demonstrates effectiveness and scalability.

## 4.9 Discussion on the Overhead of Self-MoE

One possible concern in adapting LLMs into compositional systems using Self-MoE is the potential introduction of overhead. Here, we discuss this aspect in detail, emphasizing that the additional overhead of Self-MoE is minimal while yielding significant performance gains. Essentially, the expert modules in Self-MoE are lightweight LoRA modules, contributing only about 1% additional parameters (total) for four experts, as detailed in Table 7 (Total Params). These experts are sparsely activated, which maintains low active parameters (7B + 0.3%) during inference, thus efficiently minimizing inference overhead. In contrast, traditional MoE models like Mixtral (Jiang et al., 2024) and BTX (Sukhbaatar et al., 2024) typically employ a feedforward network (FFN) layer for each expert, resulting in a significant proportional increase in total parameters as the number of experts grows, as indicated in Table 7, which demands much more memory for model loading. The design

choice in Self-MoE leads to better scalability and resource efficiency, especially when the number of experts is scaled to incorporate numerous domains of expertise.

## 5  RELATED WORK

**Combination of Experts.**   There have been numerous efforts to combine the strengths of multiple models or modules. The Mixture of Experts (MoE) models such as Switch Transformer (Fedus et al., 2022), GLAM (Du et al., 2022), and Mixtral (Jiang et al., 2024) exemplify this, dynamically allocating tasks based on the expertise of each component for better efficiency and scalability. These models contrast with ours by not prioritizing lightweight experts, resulting in a larger model with more parameters. Unlike their experts implicitly learned during pre-training, Self-MoE explicitly creates semantic experts for targeted improvements.

Another relevant area is merging, involving the weighted averaging of multiple models to form a single, aggregated model (Wortsman et al., 2022; Matena & Raffel, 2022; Ilharco et al., 2023; Jin et al., 2023). One of the leading methods, TIES (Yadav et al., 2023) tackles conflicts and parameter inconsistencies among models. DARE (Yu et al., 2024) further reduces the redundancy of parameters. However, these methods are fundamentally static in that they operate with fixed parameters once merged, which may lead to interference, lacking the dynamic flexibility that MiXSE offers.

There exist notable recent MoE models that similarly explore the utilization of semantic experts, albeit in distinct contexts (Gururangan et al., 2022; Wu et al., 2024; Muqeeth et al., 2024; Sukhbaatar et al., 2024). MOLE relies on human-labeled data, and PHATGOOSE assumes the availability of existing expert models trained by external creators and necessitates additional training for a router on the creators' side. DEMix and BTX rely on extensive pre-training, demanding significant resources, yet it as a pre-trained model holds the potential to complement our self-training approach. Unlike MOLE and PHATGOOSE, our Self-MoE framework creates experts and a router from scratch through self-improvement, while using minimal resources, as contrasted to DEMix and BTX. To offer a broader perspective, Table 7 in Appendix presents a comprehensive summary of various models that, while relevant, are not directly comparable. For further discussions and a more detailed comparison, please refer to Appendix D.1.

**Self-Improvement and Specialization of LLMs.**   The pursuit of enhancing the capabilities of LLMs often revolves around an instruction-tuning scheme, which can significantly boost cross-task generalizability (Ouyang et al., 2022; Su et al., 2022; Mishra et al., 2022; Wei et al., 2022). Due to the bottlenecks of expensive annotation costs which lead to limited scalability, the self-training concept (Luo, 2022) has gained attention from the community, where LLMs are aligned with automatically self-generated synthetic instructions (Wang et al., 2023; Sun et al., 2023; Li et al., 2024b). These are distinguished from distillation techniques (Hinton et al., 2015; Kang et al., 2023), which assume a stronger teacher model (Mitra et al., 2023; Li et al., 2024a; Sudalairaj et al., 2024), limiting their applicability.

With the growing need to adapt generalist models to specific domains, Kang et al. (2024) adopts the self-training for specialization, tackling that general instruction tuning is rarely effective in expert domains. While this work lays a foundation for enhancing specialized expertise with minimal resources, we recognize inherent trade-offs in a monolithic structure, such as performance compromises outside specialized domains. Conversely, our Self-MoE achieves uncompromising multiple expertise with a modular approach without extensive resources and adding many parameters.

## 6  CONCLUSION

In this study, we proposed Self-MoE to build compositional LLMs with self-specialized experts, MiXSE, to enhance targeted capabilities, adaptability, and interpretability without the reliance on extensive human-labeled data. Empirical evaluations across diverse domains with multiple base models demonstrated that MiXSE significantly enhances base LLM performance and overcomes specialization trade-offs. We believe this work offers a step towards modular, self-improving paradigms which can address the inherent limitations of monolithic models, providing a promising direction for future LLM research.

ACKNOWLEDGMENTS

This research is supported in part by the NSF under grant number IIS-2052498. Any opinions, findings, and conclusions or recommendations expressed in this material are those of the author(s) and do not necessarily reflect the views of the National Science Foundation.

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

## A   EXPERIMENT DETAILS

We provide each of our self-specialization prompts for knowledge, reasoning, math, and coding experts in Tables 11, 12, 13, and 14. We largely follow Kang et al. (2024)'s prompt structure to ensure quality, with additional domain-specific instructions that inform task-related information.

For our evaluation, we employ popular and widely accepted evaluation frameworks to pursue standard evaluation setups and protocols: HELM (Liang et al., 2023), LM Evaluation Harness (Gao et al., 2023), and BigCode Evaluation Harness (Ben Allal et al., 2022). We use Huggingface PEFT (Mangrulkar et al., 2022) and XLoRA (Buehler & Buehler, 2024) for the implementation of MoE compatible with LoRA.

Regarding seed instructions, we sampled 100 training instances from each of the MMLU, BBH, and GSM8K datasets, for knowledge, reasoning, and math domains, respectively. For coding, since the size of the HumanEval dataset is very small and thus the training set is not available, we took 100 samples from the MBPP training set and converted the task format to make them suit the HumanEval.

During instruction generation, we use three seed data, which are randomly sampled, as in-context examples, using a temperature of 1 and top-p of 0.98, whereas we use five seed data in-context for response generation with greedy decoding. For specialization, we use LoRA applied to all modules with a rank of 8 and alpha of 16, and train it using a learning rate of 3e-4, epochs of 3, and batch size of 32. We train each module and MiXSE using a standard Alpaca (Taori et al., 2023) prompt template on a single A100-80GB, which takes only a few hours.

## B   LIMITATIONS

While our study demonstrates promising results for the Self-MoE, we recognize areas requiring further investigation in future work. Employing self-specialization Kang et al. (2024) to generate synthetic data within our framework may raise concerns about potential data contamination and noise. Nonetheless, findings from Kang et al. (2024), which conducted an n-gram overlap analysis between the self-specialization data and test data, confirmed no significant overlap, thus alleviating the concerns about contamination. Despite this, the need for continuous monitoring of potential biases from pre-training and the development of enhanced data validation and noise filtering strategies remain important, and may present interesting direction for future work. Moreover, due to computational constraints, we did not scale our model and data to their full potential. We also did not work on the optimization of the XLoRA, the MoE module we used, to focus purely on the research problem defined in this study. Future work should therefore concentrate on overcoming these limitations, which will enable better data quality and more extensive training to unveil the full potential of the Self-MoE framework.

Table 6: Dataset statistics. Non-Target (In-Expertise) indicates where MiXSE does not directly specialize using seed data directly while relevant to targets. Non-Target (Out-of-Expertise) refers to irrelevant cases.

| Category | Benchmark | # Examples |
|---|---|---|
| *Target* | | |
| Academic Knowledge | MMLU (57 Tasks) | 14,079 |
| Reasoning | BBH (27 Tasks) | 6,511 |
| Math | GSM8K | 8,790 |
| Coding | HumanEval | 164 |
| *Non-Target (In-Expertise)* | | |
| Math | MATH | 12,500 |
| Coding | MBPP | 257 |
| *Non-Target (Out-of-Expertise)* | | |
| World Knowledge | Natural Questions | 3,610 |
| | TriviaQA | 17,200 |
| Commonsense | Hellaswag | 10,000 |
| | PIQA | 3,000 |
| Safety | TruthfulQA | 817 |

Table 7: Additional comparisons with other models for references. Results are extracted from each corresponding paper, except for pre-training methods where the numbers are all from BTX (Sukhbaatar et al., 2024).

| Method | Total Params | Active Params | Compos-itional | Semantic Experts | Light-weight | Data & Resrc -Efficient | w/o Teacher & Labels | Knowledge (MMLU 5-shot) | Reasoning (BBH) | Math (GSM8K) | Coding (HumanEval) |
|---|---|---|---|---|---|---|---|---|---|---|---|
| *Base LLM* | | | | | | | | | | | |
| Gemma 7B (Team et al., 2024) | 7B | 7B | ✗ | - | - | - | - | 65.7 | 56.1 | 42.5 | 34.1 |
| LLaMA-2 70B (Touvron et al., 2023) | 70B | 70B | ✗ | - | - | - | - | 68.9 | 51.2 | 35.2 | 29.9 |
| Mixtral 8x7B (Jiang et al., 2024) | 47B | 13B | ✓ | ✗ | ✗ | - | - | 70.6 | 67.1 | 65.7 | 32.3 |
| *Pre-training Methods* | | | | | | | | | | | |
| Branch-Train-Merge (4x7B) (Li et al., 2022) | <24B | 11.1B | ✓ | ✓ | ✗ | ✗ | ✓ | 44.3 | - | 27.7 | 30.6 |
| Sparse Upcycling (4x7B) (Komatsuzaki et al., 2023) | <24B | 11.1B | ✓ | ✓ | ✗ | ✗ | ✓ | 52.1 | - | 40.1 | 26.2 |
| Branch-Train-Mix (4x7B) (Sukhbaatar et al., 2024) | <24B | 11.1B | ✓ | ✓ | ✗ | ✗ | ✓ | 52.5 | - | 37.1 | 28.7 |
| *MoE w/ LoRA* | | | | | | | | | | | |
| PHATGOOSE (Muqeeth et al., 2024) | <4B | >3B | ✓ | ✓ | ✓ | ✗ | ✗ | - | 35.6 | - | - |
| MOLE (Wu et al., 2024) | - | - | ✓ | ✓ | ✓ | ✗ | ✗ | - | 42.2 | - | - |
| *Distillation/Synthetic Data from Larger Models* | | | | | | | | | | | |
| GLAN 7B (w/ GPT-4) (Li et al., 2024a) | 7B | 7B | ✗ | - | - | ✗ | ✗ | 62.9 | 60.7 | 80.8 | 48.8 |
| Orca-2 7B (w/ GPT-4) (Mitra et al., 2023) | 7B | 7B | ✗ | - | - | ✗ | ✗ | 53.9 | 42.8 | 55.7 | 17.1 |
| Merlinite 7B (w/ Mixtral 8x7B) (Sudalairaj et al., 2024) | 7B | 7B | ✗ | - | - | ✗ | ✗ | 64.9 | - | 44.6 | - |
| *Self-Improving* | | | | | | | | | | | |
| Ours | 7B + 1% | 7B + 0.3% | ✓ | ✓ | ✓ | ✓ | ✓ | 66.2 | 61.1 | 52.5 | 37.8 |

# C  DATASET DESCRIPTIONS

The statistics for each dataset are provided in Table 6. The target datasets used are as follows:

- **MMLU** (Massive Multitask Language Understanding) (Hendrycks et al., 2021a): A collection of 57 academic knowledge tasks.
- **BBH** (BIG-Bench Hard (Suzgun et al., 2022): A set of 27 challenging reasoning tasks.
- **GSM8K** (Grade School Math 8K) (Cobbe et al., 2021): A diverse set of grade school math word problems.
- **HumanEval** (Chen et al., 2021): A hand-written evaluation set for python programming problems.

# D  ADDITIONAL RESULTS

## D.1  ADDITIONAL COMPARISON AND DISCUSSION

In Table 7, we present additional comparisons with various other models and methods to provide a broader perspective, though comparisons may not appear to be direct, due to factors involved such as parameters, resources, etc. We discuss some noteworthy points.

Notably, although MiXSE significantly improves upon its base model, Gemma 7B, it does not yet reach the performance levels of the more powerful Mixtral 8x7B. It is important to understand that Mixtral also utilizes an MoE (Mixture of Experts) architecture, but unlike MiXSE, it does not prioritize lightweight experts, leading to a much larger model with significantly more parameters. Moreover, while Mixtral's experts are implicitly built during pre-training, MiXSE explicitly creates semantic experts, allowing for targeted improvements and clearer interpretability. Importantly, our self-improving method can be potentially applied on top of any pre-trained model including Mixtral in principle.

Similarly, BTX (Branch-Train-MiX) uses a pre-training MoE strategy where parameter-heavy semantic experts are employed, yielding substantial enhancements over the base LLM. This approach highlights the effectiveness of using semantically rich experts to refine the model's capabilities. To make comparisons in terms of efficiency, our model uses fewer parameters (7B), compared to BTX (12B active with much more whole parameters) and requires only about 1 GPU day for training, compared to 900 GPU days for BTX. In essence, since BTX is also a pre-training method while specialized, we expect it to be complementary to our Self-MoE, as evidenced in previous work (Kang et al., 2024).

With a shared spirit, MOLE and PHATGOOSE build a MoE (Mixture of Experts) using LoRA, which is semantic and lightweight. However, there are significant differences in foundational assumptions: MOLE depends on human-labeled data, while PHATGOOSE requires access to pre-

Table 8: Detailed results of Self-MoEs w/ other LLMs, comparing with each corresponding LLM and instance merging on top of it. For MMLU, we employ the 0-shot setting, based on established observations (Dettmers et al., 2023; Lin et al., 2024) that tuning yields only marginal effects in the 5-shot setting for this task. Notably, we see that any tunings improve MMLU yet still, our MiXSE demonstrates noticeable average gains over instance merging for most base models.

| Method | Knowledge (MMLU) | Reasoning (BBH) | Math (GSM8K) | Coding (HumanEval) | Avg. |
|---|---|---|---|---|---|
| *LLaMA-3 8B* | | | | | |
| Base LLM | 31.6 | 60.8 | 49.0 | 26.2 | 41.9 |
| Instance Merging | 62.5 | 46.9 | 47.5 | 24.4 | 45.3 |
| MiXSE | 61.7 | 61.5 | 52.0 | 29.3 | 51.1 |
| *Gemma 7B* | | | | | |
| Base LLM | 58.4 | 56.1 | 42.5 | 34.1 | 47.8 |
| Instance Merging | 62.6 | 57.6 | 53.5 | 36.0 | 52.4 |
| MiXSE | 65.6 | 61.1 | 52.5 | 37.8 | 54.3 |
| *LLaMA-2 7B* | | | | | |
| Base LLM | 17.8 | 38.5 | 13.0 | 12.8 | 20.5 |
| Instance Merging | 45.2 | 36.8 | 13.0 | 13.4 | 27.1 |
| MiXSE | 44.0 | 38.3 | 13.5 | 14.0 | 27.5 |
| *LLaMA-2 13B* | | | | | |
| Base LLM | 20.4 | 45.6 | 22.5 | 16.5 | 26.2 |
| Instance Merging | 51.2 | 43.0 | 25.5 | 17.1 | 34.2 |
| MiXSE | 52.1 | 45.6 | 25.0 | 17.1 | 35.0 |
| *Mistral 7B* | | | | | |
| Base LLM | 29.8 | 54.9 | 38.0 | 27.4 | 37.5 |
| Instance Merging | 61.7 | 51.5 | 30.5 | 29.2 | 43.2 |
| MiXSE | 62.0 | 58.1 | 38.0 | 28.0 | 46.5 |

trained expert models developed externally. In contrast, our Self-MoE framework independently constructs both experts and a router entirely from scratch, focusing on self-improvement without such dependencies. While their scenarios are considered reasonable in a certain context, we aim for broader applicability by minimizing assumptions on conditions.

Lastly, GLAN demonstrates outstanding performance across various domains. This is attributed to their reliance on distilling from the larger and stronger model, GPT-4, using a huge amount of data (e.g., 10 million). As outlined in our problem statement (Section 2), we deliberately avoid assuming the availability of such advanced models to ensure the broader applicability of our method which self-improves from scratch. Consequently, while acknowledging each of their own value, it is crucial to recognize that direct comparisons may not be entirely appropriate, given the fundamental differences in resource assumptions and initial conditions.

## D.2 DETAILED RESULTS OF SELF-MOE WITH OTHER BASE LLMS

Table 8 presents the detailed results of our Self-MoE applied to a diverse set of base LLMs including LLaMA-3 8B, Gemma 7B, LLaMA-2 7B and 13B, Mistral 7B. As discussed in 4.5, overall, our approach can improve base models, outperforming the strong instance merging baseline, particularly with newer/stronger models like Gemma 7B, Mistral 7B, and LLaMA-3 8B. In specific cases like LLaMA-2 for reasoning, however, we see no improvement, while improving on average. This can be attributed to the weaker baseline performance, which hinders the generation of high-quality specialized synthetic data for specific capabilities. Through manual inspection of small sample sets, we identified instances where the generated instructions exhibited poor quality, including issues such as repeated tokens, off-topic content, and other inconsistencies, not following given instructions. This highlights an opportunity for further refinement in synthetic data generation techniques, which we view as a complementary area of ongoing research. As methods for synthetic data genera-

Table 9: Results of MiXSE using only seed data. Seed Only training shows only marginal improvements over the Base LLM in some benchmarks, validating that the effect of Self-MoE is not merely due to the use of seed data.

| Benchmark | Base LLM | Seed Only | MiXSE |
|---|---|---|---|
| Knowledge (MMLU) | 58.3 | 57.4 | 65.6 |
| Reasoning (BBH) | 56.1 | 57.0 | 61.1 |
| Math (GSM8K) | 42.5 | 45.0 | 52.5 |
| Coding (HumanEval) | 34.1 | 34.1 | 37.8 |
| Avg. | 47.8 | 48.4 | 54.3 |

Table 10: Visualized examples of the token-level routing, where each token is highlighted according to assigned experts ( knowledge , reasoning , math , coding ). Different experts can be dynamically activated within an instance, as the routing operates at token-level, while the most relevant expert is primarily selected.

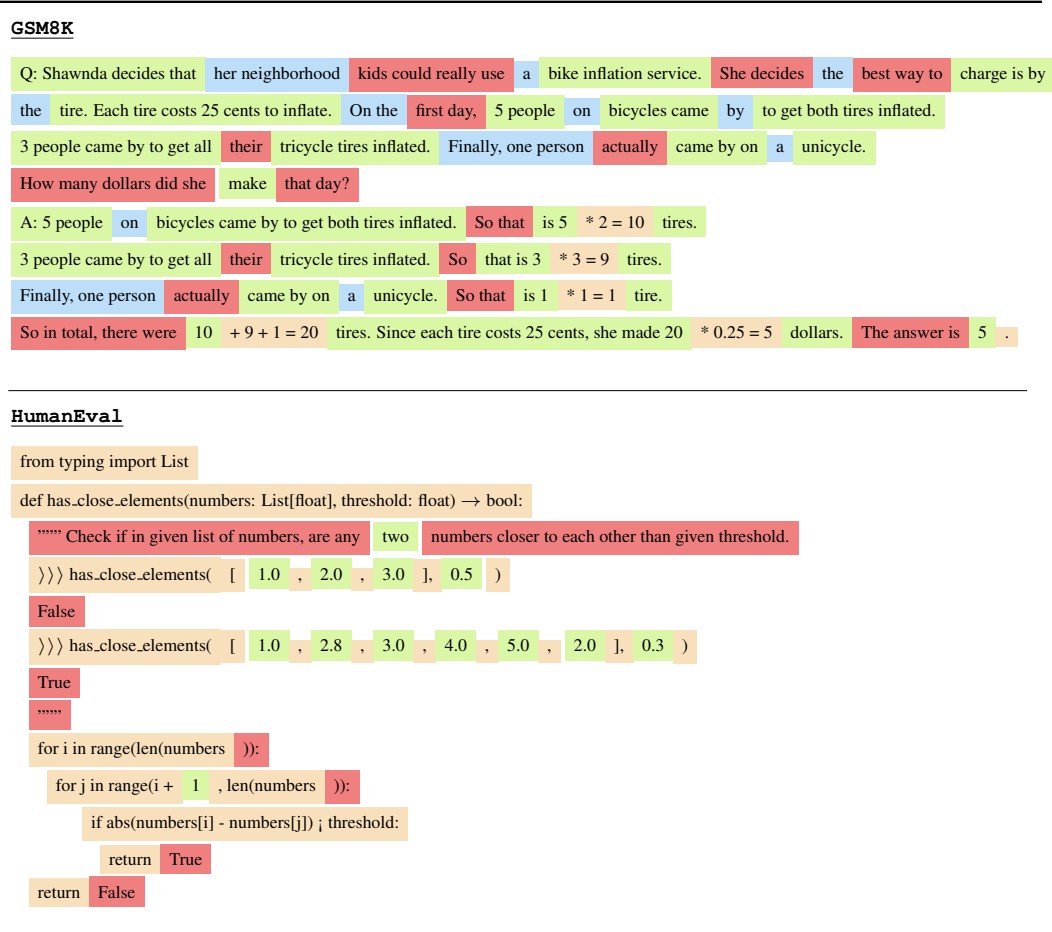

tion advance, they can directly enhance Self-MoE's performance with better self-specialized expert modules.

## D.3 MIXSE USING ONLY SEED DATA

Table 9 shows the results of the MiXSE when exploiting only seed data for training, clarifying the benefits derived from our methodological enhancements beyond the mere inclusion of seed data in training. While the Seed Only shows slight improvements over the Base LLM in some benchmarks, the significant enhancements of our MiXSE across all benchmarks confirm that the enhanced capabilities of Self-MoE are not merely due to the use of seed data. This further highlights the achievement of self-improvement with our method.

## D.4 VAILDITY OF COMPARATIVE RESULTS

In an effort to address the concern related to the sensitivity of in-context learning (Min et al., 2022), we perform three runs with the different lists of few-shot samples where applicable. As a result, we see that the mean of the base LLM (Gemma-7B)'s average performance across domains is 47.9 with a standard deviation (SD) of 0.56, that of our MiXSE is 53.6 with an SD of 0.60, and that of instance merging is 51.6 with an SD of 0.87. A statistical analysis between MiXSE and instance merging yields a p-value of 0.03, confirming the significant difference.

## D.5 VISUALIZED EXAMPLES OF ROUTING DECISION

Table 10 provides a detailed visualization of token-level routing decisions based on the Top-1 selection configuration. This table highlights how the routing module dynamically activates different experts within a single instance, reflecting the flexibility of token-level operation. As illustrated, the most relevant expert is predominantly selected for each token; however, the system occasionally activates other experts dynamically, depending on the specific token context within the instance. This behavior contrasts with self-specialization, which consistently relies on a single expert to handle all tokens uniformly, lacking the token-level granularity observed in the routing mechanism.

Table 11: Prompts for knowledge-related instruction and response generation.

**Instruction Brainstorming Prompt**

```
You are asked to come up with a set of task instructions about diverse domains across STEM,
humanities, social sciences, and others.  These task instructions will be given to a language
model and we will evaluate the model for completing the instructions.

Here are the requirements:
1.  The type of task should be multiple-choice question answering.  That is, a question along
with multiple options (A, B, C, D) should be provided.
2.  The language used for the instruction/question also should be diverse.
3.  A language model should be able to complete the instruction.  For example, do not ask the
assistant to create any visual or audio output.  For another example, do not ask the assistant
to wake you up at 5pm or set a reminder because it cannot perform any action.
4.  The instructions should be in English.
5.  The instructions should be 1 to 2 sentences long.  Either an imperative sentence or a
question is permitted.
6.  You should generate an appropriate input to the instruction.  The input field should
contain a specific example provided for the instruction.  It should involve realistic data and
should not contain simple placeholders.  The input should provide substantial content to make
the instruction challenging.
7.  Ensure diverse domains are covered for extensive expert-level knowledge.  The subjects
may include Abstract Algebra, Anatomy, Astronomy, Business Ethics, Clinical Knowledge,
College-level Biology, Chemistry, Computer Science, Mathematics, Medicine, Physics, Computer
Security, Conceptual Physics, Econometrics, Electrical Engineering, Elementary Mathematics,
Formal Logic, Global Facts, High School-level Biology, Chemistry, Computer Science, European
History, Geography, Gov't and Politics, Macroeconomics, Mathematics, Microeconomics, Physics,
Psychology, Statistics, US History, World History, Human Aging, Human Sexuality, International
Law, Jurisprudence, Logical Fallacies, Machine Learning, Management, Marketing, Medical
Genetics, Miscellaneous, Moral Disputes, Moral Scenarios, Nutrition, Philosophy, Prehistory,
Professional-level (Accounting, Law, Medicine, Psychology), Public Relations, Security
Studies, Sociology, US Foreign Policy, Virology, World Religions, etc.

List of tasks:
```

**Response Generation**

```
You are a knowledgeable domain expert.  Given an instruction and a question, generate the
best answer to solve the given task about STEM, humanities, social sciences, and others.
```

Table 12: Prompts for reasoning-related instruction and response generation.

**Instruction Brainstorming Prompt**

```
You are asked to come up with a set of task instructions focusing on challenging tasks that
require multi-step reasoning.  These task instructions will be given to a language model and
we will evaluate the model for completing the instructions.

Here are the requirements:
1.  The type of task should be question answering, requiring multi-step reasoning.
2.  The language used for the instruction/question also should be diverse.
3.  The generated problem should have a single correct answer.
4.  The instructions should be in English.
5.  The instructions should be 1 to 2 sentences long.  Either an imperative sentence or a
question is permitted.
6.  You should generate an appropriate input question to the instruction.  It should
involve realistic data and should not contain simple placeholders.  The input should provide
substantial content to make the instruction challenging.
7.  Ensure diverse topics and levels are covered for extensive expert-level reasoning.  The
tasks may be about boolean expression, causal judgement, date understanding, disambiguation
of question, closing Dyck-n words, formal fallacies, geometric shapes, hyperbaton, logical
deduction of objects, movie recommendation, multi-step arithmetic problem, navigation, object
counting, table reasoning, reasoning about colored objects, selecting one that ruins the name
in an input, salient translation error detection, sarcastic sentence classification, sports
understanding, temporal sequences, tracking shuffled objects, web of lies, word sorting, etc.

List of tasks:
```

**Response Generation**

```
You are a multi-step reasoning expert.  Given an instruction and a challenging question,
generate step-by-step reasoning and the answer.
```

Table 13: Prompts for math-related instruction and response generation.

**Instruction Brainstorming Prompt**

```
You are asked to come up with a set of task instructions focusing on mathematical problems.
These task instructions will be given to a language model and we will evaluate the model for
completing the instructions.

Here are the requirements:
1.  The type of task should be question answering, requiring multi-step reasoning.
2.  The language used for the instruction/question also should be diverse.
3.  The generated mathematical problem should have a solution.
4.  The instructions should be in English.
5.  The instructions should be 1 to 2 sentences long.  Either an imperative sentence or a
question is permitted.
6.  You should generate an appropriate input question to the instruction.  It should
involve realistic data and should not contain simple placeholders.  The input should provide
substantial content to make the instruction challenging.
7.  Ensure diverse topics and levels are covered for extensive expert-level reasoning.  The
subjects may include Algebra, Counting, Probability, Calculus, Statistics, Geometry, Linear
Algebra, Number Theory and grade school math, etc.

List of tasks:
```

**Response Generation**

```
You are a math expert.  Given an instruction and a mathematical question, generate
step-by-step reasoning and the answer.
```

Table 14: Prompts for coding-related instruction and response generation.

**Instruction Brainstorming Prompt**

```
You are asked to come up with a set of task instructions focusing on coding problems.
These task instructions will be given to a language model and we will evaluate the model
for completing the instructions.

Here are the requirements:
1.  The type of task should be about coding problems, such as writing a python function given
a specific instruction and test examples.
2.  The language used for the instruction should be diverse, but the programming language
should be python.
3.  The generated problem should have a solution.
4.  The instructions should be in English.
5.  You should generate appropriate and correct test examples for the given problem.
6.  Ensure diverse functions and levels are covered for extensive expert-level coding.

List of tasks:
```

**Response Generation**

```
You are a coding expert.  Given an instruction and test cases, write a python function that
passes the test cases.
```

