# OpenReview forum: "Self-MoE: Towards Compositional Large Language Models with Self-Specialized Experts"
_ICLR.cc/2025/Conference — ICLR 2025 Poster_

### Official Review · Reviewer_fc1R · 2024-10-27

**Soundness:** 3
**Presentation:** 3
**Contribution:** 2
**Rating:** 6
**Confidence:** 4

**Summary:**

This paper proposes a self-specification method, named Self-MoE, to transform a monolithic LLM to a MoE system with self-specific experts. To construct expert modules for specific domains, this paper utilizes self-generated data from the base LLM. The resulted Self-MoE demonstrates substantial improvements over the base LLM, even achieves better performance than data mixture and weight merging.

**Strengths:**

1. It's a good idea to stimulate diverse domain-specific competencies inherent within the base model and reassemble these specific experts to achieve a more powerful MoE system.
2. This approach offers the advantage of eliminating the necessity for domain-specific data acquisition.
3. The writing and presentation are clear and easy to read.

**Weaknesses:**

1. The primary contribution of this paper is constructing a MoE system from multiple domain experts, which are fine-tuned on the domain-specific synthesis data from base model itself. However, this self-specialization method was originally proposed by [1]. Moreover, recent studies already propose the construction of MoE system based on multiple domain-specific dense models, which can be fine-tuned using LoRA [2] or full-parameter tuning [3].

2. A critical question in utilizing self-synthesized data for self-improvement is where the performance improvements comes from. For instance, the improvements in [4] is based on an additional reward model to differentiate between good and bad samples. However, the underlying mechanism of Self-MoE's self-improvement remains ambiguous. It is not immediately apparent why domain-specific datasets generated by the base model, without guidance from a stronger model or reward model, can lead to overall performance gains.

3. The expert modules in Self-MoE are fine-tuned from self-synthesized domain data, which determines the upper bound of the expert's capabilities. Therefore, more analysis of the self-synthesized datasets is needed, e.g., data diversity, complexity, and the model performance compared to using existing open-source datasets.

[1] Self-specialization: Uncovering latent expertise within large language models.

[2] Mixture of lora experts.

[3] Branch-train-mix: Mixing expert llms into a mixture-of-experts llm.

[4] Direct Nash optimization: Teaching language models to self-improve with general preferences.

**Questions:**

1. In the instruction brainstorming stage, how to generate diverse instructions within a given domain? Since the generator is not a strong instruction-following model, e.g. GPT-4, using self-instruct approach can lead to low diversity in the generated domain-specific datasets as illustrated in [5].

2. In the mixture of self-specialized experts stage, router network parameters are shared across all layers. However, when router networks in different MoE layers do not share parameters, they typically exhibit distinct routing behaviors. What is the influence on overall performance and routing behaviors if separate, layer-specific router networks are employed instead of shared ones?

3. In the main results, the baseline methods involve fine-tuning both the LoRA and router parameters. However, this approach presupposes that all base models should be implemented using the MoE architecture, potentially introducing additional complexities in router training. To ensure a more comprehensive and equitable comparison, it is essential to add additional baseline methods, which are directly fine-tuned on the base model using different datasets (specific capabilities, mixture of capabilities, model merging).

4. In Section 4.5, can the author provide the instance merging results (using all the 20k data to direct fine-tune the base model with LoRA) for different base models?

[5] Explore-Instruct: Enhancing Domain-Specific Instruction Coverage through Active Exploration

I would be happy to discuss further with the authors and reassess my score based on the rebuttal stage.

---

> ### Author Response · Authors · 2024-11-21
> **Response to Reviewer fc1R (Part 1)**
>
> We appreciate the reviewer’s thoughtful comments and feedback on our paper. We are encouraged to see that you are open to further discussion and reassessment.
>
> ___
>
> > *self-specialization method was originally proposed by [1]. Moreover, recent studies already propose the construction of MoE system …*
>
> It is true that self-specialization was previously proposed and we use it as part of our framework. Self-MoE builds on self-specialization, and targets mitigating the issue of forgetting (performance reduction for non-specialized tasks) inherent to [1] as indicated in Figure 1 and Table 1. Our work differs from prior work on MoE, which relies on human-labeled data for specialization, or assumes the existence of trained modules [2]. Instead, Self-MoE constructs individual lightweight expert modules from scratch using self-generated data similar to [3].  As far as we are aware, this is the first work to use self-specialized experts which are trained in a similar manner to self-instruct in combination with MoE.
>
> [1] Self-specialization: Uncovering latent expertise within large language models
>
> [2] Learning to route among specialized experts for zero-shot generalization
>
> [3] Self-Instruct: Aligning Language Models with Self-Generated Instructions
>
> ___
>
> > *A critical question in utilizing self-synthesized data for self-improvement is where the performance improvements comes from.*
>
> We understand the reviewer’s question regarding self-improvement without external models, and admit that this should better be clarified in the paper. As hypothesized in [1], “Expertise in various domains is mixed and latent within base LLMs”. Self-specialization helps uncover this latent expertise and improves domain performance without relying on teacher models. However, this comes at the cost of reducing the performance of other domains, which is fixed with our Self-MoE approach.
>
> [1] Self-specialization: Uncovering latent expertise within large language models

---

> ### Author Response · Authors · 2024-11-21
> **Response to Reviewer fc1R (Part 2)**
>
> > *more analysis of the self-synthesized datasets is needed…*
>
> Thank you for this suggestion. We conducted extensive analyses of the self-synthesized datasets, focusing on diversity, complexity, and the model performance compared to human-labeled datasets, as suggested. The findings are summarized below:
>
> First, we analyzed Type-to-Token Ratio (TTR), and semantic similarity of the pairwise instructions' embeddings using SBERT, as used in [1] for diversity measurement.
>
> | TTR (↑) | MMLU | BBH | GSM8K | HumanEval | Avg. |
> |------------------------------------|--------|-------|-------|--------|-------|
> | Human-Labeled Data | 0.2671 | 0.1672 | 0.1683 | 0.1121 | 0.1787 |
> | Synthetic Data           | 0.2639 | 0.1889 | 0.1484 | 0.0961 | 0.1743 |
>
> Synthetic data demonstrates comparable linguistic diversity to human-labeled data, with a slightly higher TTR for BBH, suggesting that the synthetic data includes richer lexical variation, especially in reasoning tasks. This indicates that the synthetic data contains a wider vocabulary or more diverse terms.
>
> | Semantic Similarity (↓) | MMLU | BBH | GSM8K | HumanEval | Avg. |
> |------------------------------------|--------|-------|-------|--------|-------|
> | Human-Labeled Data | 0.2625 | 0.1757 | 0.4125 | 0.4608 | 0.3279 |
> | Synthetic Data | 0.3129 | 0.1948 | 0.3360 | 0.4791 | 0.3307 |
>
> For semantic similarity, synthetic data achieves generally low similarity in absolute terms within each dataset, similar to human-labeled data (0.3307 vs. 0.3279). This suggests high semantic diversity overall, reflecting the natural diversity found in human-labeled data.
>
> Next, we leveraged a strong instruction-following model, GPT-4, as a judge to classify which instruction was synthetic. Given a pair of human-labeled and synthetic instructions, the classification accuracy was as follows:
>
> | Classification Accuracy (↓) | MMLU | BBH | GSM8K | HumanEval |
> |----------------------|--------|-------|-------|--------|
> | LLM as-a-judge | 55% | 68% | 60% | 50% |
>
> The classification accuracy ranged from 50% (random guessing) to 68%, with the lowest accuracy for HumanEval and MMLU, suggesting that synthetic data closely mimics human complexity in these domains. Conversely, the higher accuracy for BBH and GSM8K indicates that synthetic data in these domains has room to improve.
>
> Finally, we compared the performance of Self-MoE fine-tuned with synthetic data against human-labeled seed data. The results are summarized below:
>
> | Configuration | Knowledge | Reasoning | Math | Coding | Avg. |
> |------------------------------------|--------|-------|-------|--------|-------|
> | w/ Human-labeled data (Seed) | 57.4 | 57.0 | 45.0 | 34.1 | 48.4 |
> | w/ Synthetic data (1x)               | 57.7 | 55.9 | 45.5 | 32.9 | 48.0 |
> | w/ More Synthetic data (5x)      | 61.3 | 58.4 | 48.4 | 36.6 | 51.2 |
> | w/ More Synthetic data (50x)    | 65.6 | 61.1 | 52.5 | 37.8 | 54.3 |
>
> We observed that with the same quantity (400) as the seed, synthetic data achieves performance similar to human-labeled data in average. When scaling up the size (5x and 50x), synthetic data demonstrates the effectiveness and scalability of synthetic data in fine-tuning expert modules.
>
> We will include these analyses in the final paper to provide a comprehensive evaluation of synthetic data. Thank you for your valuable feedback!
>
> [1] Improving Diversity of Commonsense Generation by Large Language Models via In-Context Learning
>
> ___
>
> > *how to generate diverse instructions within a given domain? Since the generator is not a strong instruction-following model, e.g. GPT-4, using self-instruct approach can lead to low diversity*
>
> We may not expect the synthetic data generated by base 7B models to be as good as the ones by GPT-4. However, in order to diversify the instructions as much as possible, the in-context seed demonstrations are randomly sampled/shuffled and incorporated into the prompt, and instructions are generated through sampling with a high temperature to promote variety.
>
> It is important to note that the goal of generating these instructions is not necessarily to teach the model new skills, but rather to help shift the model’s specialization towards specific domains. Our experiments show that the synthetic data generated is sufficiently effective for this purpose. It enables the model to focus on its existing knowledge and skills, potentially at the expense of other knowledge. This self-produced data has been empirically observed to be effective in driving the model toward better specialization within specific domains.
>
> We believe that synthetic data diversification is an important, ongoing area of research that complements our Self-MoE approach but remains orthogonal to it. As methods for synthetic data generation advance, they can directly enhance Self-MoE’s performance with better self-specialized expert modules.

---

> ### Author Response · Authors · 2024-11-21
> **Response to Reviewer fc1R (Part 3)**
>
> > *What is the influence on overall performance and routing behaviors if separate, layer-specific router networks are employed instead of shared ones?*
>
> Thank you for raising this thoughtful question. To investigate the impact of separate, layer-specific routers, we have conducted experiments comparing their performance to the shared router. The results are summarized below:
>
> | Configuration | Router Params | Knowledge | Reasoning | Math | Coding | Avg. |
> |---------------------------------------------------|--------|--------|-------|-------|--------|-------|
> | Base LLM                                            | 0       | 58.4 | 56.1 | 42.5 | 34.1 | 47.8 |
> | Self-MoE (w/ Shared Router)              | 1x     | 65.6 | 61.1 | 52.5 | 37.8 | 54.3 |
> | Self-MoE (w/ Layer-Specific Routers) | 200x | 59.5 | 59.1 | 50.5 | 32.9 | 50.5 |
>
> We found that the shared router achieves an average score of 54.3, significantly outperforming the layer-specific router’s 50.5, despite the latter having about 200x more parameters. We conjecture this might be attributed to the fact that the shared one enforces a consistent routing mechanism across layers, which likely helps maintain coherence in expert activation, whereas the layer-specific ones may introduce conflicting routing decisions, possibly requiring more data or hyperparameter tuning to become effective. This is an initial result, which requires further investigation on this in future work. We will include these findings and discussions in the final version of the paper. Thank you for bringing up this important point!
>
> ___
>
> > *it is essential to add additional baseline methods, which are directly fine-tuned on the base model using different datasets (specific capabilities, mixture of capabilities, model merging).*
>
> Thank you for your suggestion. We want to clarify that the main results section (Table 1) already includes the baseline comparisons suggested:
> 1) Direct fine-tuning on specific capabilities: We trained individual versions of the base model on domain-specific datasets, allowing each model to self-specialize in a single capability (e.g., knowledge, reasoning). The strongest one among these baselines appeared to be a math-specialized expert, which however led to a 3.2 drop in average compared to our method.
> 2) Fine-tuning on a mixture of capabilities: We also included a multitask setup (i.e., instance merging), where the base model was fine-tuned on a combined dataset that covers multiple domains. This served as the strongest baseline, leading to a 1.9 drop in average performance compared to our Self-MoE.
> 3) Model merging: We included weight-merging methods that combine the strengths of multiple models specialized for different capabilities without using MoE’s dynamic routing. This baseline resulted in a 6.4 decrease in performance, compared to ours.
>
> ___
>
> > *can the author provide the instance merging results (using all the 20k data to direct fine-tune the base model with LoRA) for different base models?*
>
> Thank you for this suggestion. We agree that adding instance merging for different base models provides a meaningful and comparative baseline. We have conducted experiments comparing Self-MoE with instance merging and each base LLM across different base models, as summarized below:
>
> |                        | Base LLM | Instance Merging | Self-MoE |
> |-----------------|-------|-------|-------|
> | LLaMA-2 7B   | 20.5 | 27.1 | 27.5 |
> | LLaMA-2 13B | 26.2 | 34.2 | 35.0 |
> | Mistral 7B       | 37.5 | 43.2 | 46.5 |
> | LLaMA-3 8B   | 41.9 | 45.3 | 51.1 |
> | Gemma 7B     | 47.8 | 52.4 | 54.3 |
>
> We observed instance merging demonstrates meaningful improvement over each base LLM, providing a strong baseline method for comparison. However, Self-MoE consistently outperforms instance merging across all tested base models, especially with newer or stronger models like Mistral 7B, LLaMA-3 8B, and Gemma 7B. This highlights Self-MoE’s ability to dynamically activate relevant experts, avoiding the interference that can arise from uniformly fine-tuning on all tasks. We will incorporate these results into the final version to provide a more comprehensive evaluation in Section 4.5 and Figure 4. Thank you again for this insightful suggestion!

---

> > ### Comment · Reviewer_fc1R · 2024-11-22
> > **Further Questions from Reviewer fc1R**
> >
> > > more analysis of the self-synthesized datasets is needed…
> >
> > Contrary to the authors' assertions, the semantic similarity of synthetic data is higher than that of human-labeled data in three out of four datasets. For instance, in the MMLU dataset, the semantic similarity of synthetic data (0.3129) is higher than that of human-labeled data (0.2625), indicating lower diversity of the synthetic data.
> >
> > > it is essential to add additional baseline methods, which are directly fine-tuned on the base model using different datasets (specific capabilities, mixture of capabilities, model merging).
> >
> > It is worth noting that Table 1 presents the active parameters of these baseline methods as 7B + 0.3%, which is equivalent to the proposed Self-MoE method. However, the authors suggest that these baselines are directly fine-tuned on the base model, utilizing 7B active parameters. This apparent inconsistency merits clarification and further investigation.

---

> > > ### Author Response · Authors · 2024-11-22
> > > **Thank you for your questions!**
> > >
> > > Thank you for your additional questions! We would like to clarify a few points below.
> > >
> > > > *the semantic similarity of synthetic data is higher than that of human-labeled data in three out of four datasets.*
> > >
> > > Sorry for the confusion. “This suggests **high**$\textcolor{red}{\text{er}}$ semantic diversity” was a typo. We meant to say “high (in general)”, not “higher” in this sentence.
> > >
> > > We agree that the semantic similarity of synthetic data is indeed higher in three out of the four datasets compared to human-labeled data. We simply meant to convey that the semantic similarity of synthetic data is generally low in absolute terms, indicating good diversity overall. On average, the semantic similarity between synthetic and human-labeled data is quite similar, which we believe is a promising result. We apologize for any confusion caused by our previous response, and we appreciate your careful consideration of this matter. We have updated our paper to include these analyses in Section 4.8.
> > >
> > > > *However, this approach presupposes that all base models should be implemented using the MoE architecture, potentially introducing additional complexities in router training. To ensure a more comprehensive and equitable comparison, it is essential to add additional baseline methods*
> > >
> > > Thank you for pointing this out. We realize there may have been a miscommunication in our previous discussion. To clarify, **Self-MoE does not require the base models to be implemented using the MoE architecture**. Rather, Self-MoE builds upon purely any base LLMs using LoRA-based fine-tuning like other baselines, but the base model itself does not need to be MoE-based. All the approaches discussed in our paper, Self-MoE and the baselines, involve parameter-efficient fine-tuning without requiring a full MoE setup. They all use the **same active parameters** at inference time, which ensures that the comparisons are **fair** and consistent. We apologize for any confusion caused. To clarify, we have updated the paper to provide a clearer explanation on this in the Baselines Section.
> > >
> > > We greatly appreciate your valuable feedback and consideration!

---

> > > > ### Comment · Reviewer_fc1R · 2024-11-23
> > > > **Response from Reviewer fc1R**
> > > >
> > > > I found that my concerns have been addressed. I have adjusted my rate accordingly.

---

> > > > > ### Author Response · Authors · 2024-11-23
> > > > >
> > > > > Thank you for raising your score. We are glad to hear that your concerns have been addressed. Your valuable insights and feedback have been instrumental in refining our paper!

---

### Official Review · Reviewer_eqDV · 2024-11-02

**Soundness:** 3
**Presentation:** 3
**Contribution:** 3
**Rating:** 6
**Confidence:** 4

**Summary:**

This paper proposes a new paradigm of self-MOE that designs a compositional LLM integrating various experts specialized in different aspects and employing a trained router to dynamically choose experts during inference. The experts are trained with LoRA and only a small proportion of parameters are adjusted during self-specialization, keeping a high efficiency. Self-MOE has been tested on several popular LLMs, e.g., Gemma and LLaMA 3, proving its strong ability to enhance current LLMs' performance when dealing with tasks in various aspects simultaneously.

**Strengths:**

1. Novel idea of self-MOE and it can be a promising new paradigm for stronger foundation models.
2. Paper is well-written.

**Weaknesses:**

1. Some claims are not very intuitive. I have questions about how synergies among expert happen and given the synergies, why only 1 expert is activated. Please refer to the questions below.
2. Some parts in the main body of this paper are not necessary, for example Table 5. I would rather see it in the appendix. Some figures are either too huge or loosely-structured and leave lots of wasted blank space (e.g., Figure 1 and 2). I think you can put more experiments about self-MOE in the main body. Please refer to point 2 of the questions below. And I would also prefer to see more detailed analysis about other base LLMs. Currenlty I only see a very abbreviated figure (Figure 4) related to this. I think breaking down their performances into several parts as how you present Gemma 7B is meaningful.

I actually like this paper. Please kindly address my concerns and questions. I think a good rebuttal will make me change my mind and raise the score. Good luck with the rebuttal.

**Questions:**

1. Line 223. I am just wondering if the top-k operation is after doing softmax, then the probability of expert selection is not normalized, meaning that when k is small alpha will also be small. Do you think it will lead to the experimental results which show that top-1 self-MOE is better than top-2 self-MOE (as presented in Table 2). I think by not normalizing the selection probability, the amount of information introduced by experts will be different when different numbers of experts are activated. So directly comparing top-1 and top-2 self-MOE is a bit unfair. Maybe the reason for the worse performance of top-2 self-MOE comes from the excessive redundant information introduced by an irrelavant expert. But this guess contradicts your claim in line 88 that MiXSE explores synergies among experts. I wish to have a more detailed discussion about this part. Given the hypothesis that synergies among experts help to boost performance, why top-1 self-MOE outperforms top-2 self-MOE? This is important for this work, because by only choosing one expert, it largely resembles self-specialization.
2. Line 348. I am also suspicious about the claim that jointly training router and experts would make semantic distinctions among experts diluted. Have you tried to jointly train them and activate more experts (>= 2)? Could you provide it in your rebuttal (sorry I know it requires lots of computational resources)? I just think the experiments and the obervations are not fully supported by your claim. This is also related to your claim that synergies among experts happen when you have multiple experts controlled by a router in self-MOE. If you don’t jointly train them, how synergies happen?
3. Again in Section 4.3, the routing distribution shows that datasets like MMLU benefit from both knowledge and reasoning experts. So why don’t you just activate more experts?
4. Section 4.7. Have you tried adding experts in different orders? For example, R + M + C + K.
5. In Table 3, you labeled TruthfulQA as safety. I am not sure why.

---

> ### Author Response · Authors · 2024-11-21
> **Response to Reviewer eqDV (Part 1)**
>
> Thank you for your thoughtful comments and questions. We are glad to hear that you like our paper. We appreciate the opportunity to clarify key points.
>
> ___
>
> > *I have questions about how synergies among experts happen and given the synergies, why only 1 expert is activated …*
>
> > *Again in Section 4.3, the routing distribution shows that datasets like MMLU benefit from both knowledge and reasoning experts. So why don’t you just activate more experts?*
>
> In our experiment, we observed that **top-1 and top-2 routing were similarly the best** in overall performance, while activating all experts decreases the performance. This suggests that identifying and leveraging the most relevant expert for a given task is typically sufficient and most effective. Given this, we opted for the top-1 configuration (sparser activation) for better efficiency.
>
> Regarding synergies among experts, the synergies stem from the **token-wise** application of the routing mechanism. Please refer to our response below.
>
> ___
>
> > *Given the hypothesis that synergies among experts help to boost performance, why top-1 self-MOE outperforms top-2 self-MOE? This is important for this work, because by only choosing one expert, it largely resembles self-specialization.*
>
> Thank you for your insightful and thorough question. Let us clarify about the synergies among experts. Our MoE framework operates at the **token level**, meaning that, even with a top-1 selection, **different experts can be dynamically activated within a single task or instance**, allowing for synergies across tokens. **Therefore, even with top-1 expert selection, Self-MoE diverges from self-specialization.** When more experts are selected per token, this can introduce unnecessary overlap, leading to performance degradation due to less targeted information, as can be seen in Table 2 (top1/2 vs. Top-All).

---

> ### Author Response · Authors · 2024-11-21
> **Response to Reviewer eqDV (Part 2)**
>
> > *Line 348. I am also suspicious about the claim that jointly training router and experts would make semantic distinctions among experts diluted. Have you tried to jointly train them and activate more experts (>= 2)? Could you provide it in your rebuttal (sorry I know it requires lots of computational resources)? If you don’t jointly train them, how synergies happen?*
>
> Jointly training both the router and experts could indeed dilute the semantic distinctions among experts, as it introduces dependencies that prevent each expert from specializing fully in its target domain. Our result in Table 2 shows a significant decrease in performance with joint training, indicating that it weakens domain specialization, which is one of the important findings in our work. As we have discussed above, the routing in our Self-MoE operates at the **token-level** within a task instance, and hence the synergies happen even without the joint training that actually appeared to exhibit a negative impact on specialization, rather than synergy.
>
> To address your suggestion, we conducted an additional experiment with jointly trained routers and 2 experts activated, summarized below:
>
> | Configuration | Knowledge | Reasoning | Math | Coding | Avg. |
> |-------------------------------------------|--------|-------|-------|--------|-------|
> | w/ Joint Training (Top-1) | 64.5 | 58.1 | 46.0 | 33.5 | 50.5 |
> | w/ Joint Training (Top-2) | 64.2 | 53.3 | 48.5 | 36.5 | 50.6 |
> | w/o Joint Training (Top-1) | 65.6 | 61.1 | 52.5 | 37.8 | 54.3 |
> | w/o Joint Training (Top-2) | 65.5 | 60.9 | 52.5 | 38.4 | 54.3 |
>
> Overall, top-1 and top-2 activation configurations exhibit similar average performance regardless of joint training. However, joint training reduces performance for both configurations, emphasizing the importance of maintaining independent specialization of experts. **With specialized experts preserving each semantic, even the more efficient top-1 activation allows for synergies among experts through dynamic token-wise routing.**
>
> ___
>
> > *Some parts in the main body of this paper are not necessary, for example Table 5. I think you can put more experiments about self-MOE in the main body.*
>
> We appreciate your suggestion and will update the paper with the proposed changes for the final version. Regarding more experiments in the main body, we agree it would greatly improve our paper, and thus we will incorporate the following results produced following reviewers' comments:
> - Joint training with top-2 activation
> - Comparison between layer-specific routers and a shared router
> - Instance merging for different base models
> - Analyses of the self-generated data compared with human-labeled data
>
> ___
>
> > *Section 4.7. Have you tried adding experts in different orders? For example, R + M + C + K.*
>
> This is an interesting question. We currently consider that the order of adding experts does not matter, as each expert’s contribution depends solely on its relevance to the token-level routing decision rather than a predefined sequence. As illustrated in the equation below (lines 216-222 in the paper), $\alpha$ represents the relevance weights computed by the router, which determines which experts to activate based on the input:
>
> $h = \theta_0 x + \sum_{i=1}^n {\alpha_i \Delta\theta_i x} $
>
> Since the router is trained starting from uniform initialization and its operation above is independent of expert order, the sequence in which experts are added does not affect the routing decision.
>
> ___
>
> > *In Table 3, you labeled TruthfulQA as safety. I am not sure why.*
>
> Thank you for pointing this out. Labeling TruthfulQA under “safety” aimed to categorize it as a benchmark focused on minimizing misinformation and maximizing model truthfulness, which we interpret as related to safety, as discussed in safety-related literature [1, 2]. However, if you believe this categorization may not be appropriate and have a better idea, we would be happy to change it!
>
> [1] SafetyPrompts: a Systematic Review of Open Datasets for Evaluating and Improving Large Language Model Safety
>
> [2] Beyond Perplexity: Multi-dimensional Safety Evaluation of LLM Compression

---

> ### Comment · Reviewer_eqDV · 2024-11-21
> **Further Question**
>
> > different experts can be dynamically activated within a single task or instance,
>
> Could you please provide an example based on this to explain the difference between self-MoE and self-specialization?
>
> I read the rebuttal, and I wish to see the updated pdf before I make the final decision.

---

> > ### Author Response · Authors · 2024-11-22
> > **Thank you for your question!**
> >
> > Sure! Please consider the following examples, where tokens are highlighted according to assigned experts ($ \small{\colorbox{lightblue}{knowledge}, \colorbox{LightCoral}{reasoning}, \colorbox{lightgreen}{math}, \colorbox{orange}{coding}} $):
> >
> > GSM8K:
> >
> > $ \small{\colorbox{lightgreen}{Q: Shawnda decides that} \colorbox{lightblue}{her neighborhood} \colorbox{LightCoral}{kids could really use} \colorbox{lightblue}{a} \colorbox{lightgreen}{bike inflation service.}} $
> > $ \small{\colorbox{LightCoral}{She decides} \colorbox{lightblue}{the} \colorbox{LightCoral}{best way to} \colorbox{lightgreen}{charge is by} \colorbox{lightblue}{the} \colorbox{lightgreen}{tire. Each tire costs 25 cents to inflate.}} $
> > $ \small{\colorbox{lightblue}{On the} \colorbox{LightCoral}{first day,} \colorbox{lightgreen}{5 people} \colorbox{lightblue}{on} \colorbox{lightgreen}{bicycles came} \colorbox{lightblue}{by} \colorbox{lightgreen}{to get both tires inflated.} \colorbox{lightgreen}{3 people came by to get all} \colorbox{LightCoral}{their} \colorbox{lightgreen}{tricycle tires inflated.}} $
> > $ \small{\colorbox{lightblue}{Finally, one person} \colorbox{LightCoral}{actually} \colorbox{lightgreen}{came by on} \colorbox{lightblue}{a} \colorbox{lightgreen}{unicycle.} \colorbox{LightCoral}{How many dollars did she} \colorbox{lightgreen}{make} \colorbox{LightCoral}{that day?}} $
> > $ \small{\colorbox{lightgreen}{A: 5 people} \colorbox{lightblue}{on} \colorbox{lightgreen}{bicycles came by to get both tires inflated.} \colorbox{LightCoral}{So that} \colorbox{lightgreen}{is 5} \colorbox{orange}{* 2 = 10} \colorbox{lightgreen}{tires.}} $
> > $ \small{\colorbox{lightgreen}{3 people came by to get all} \colorbox{LightCoral}{their} \colorbox{lightgreen}{tricycle tires inflated.} \colorbox{LightCoral}{So} \colorbox{lightgreen}{that is 3} \colorbox{orange}{* 3 = 9} \colorbox{lightgreen}{tires.}} $
> > $ \small{\colorbox{lightblue}{Finally, one person} \colorbox{LightCoral}{actually} \colorbox{lightgreen}{came by on} \colorbox{lightblue}{a} \colorbox{lightgreen}{unicycle.} \colorbox{LightCoral}{So that} \colorbox{lightgreen}{is 1} \colorbox{orange}{* 1 = 1} \colorbox{lightgreen}{tire.}} $
> > $ \small{\colorbox{LightCoral}{So in total, there were} \colorbox{lightgreen}{10} \colorbox{orange}{+ 9 + 1 = 20} \colorbox{lightgreen}{tires. Since each tire costs 25 cents, she made 20} \colorbox{orange}{* 0.25 = 5} \colorbox{lightgreen}{dollars.} \colorbox{LightCoral}{The answer is} \colorbox{lightgreen}{5} \colorbox{orange}{.}} $
> >
> > HumanEval:
> >
> > $ \small{\colorbox{orange}{from typing import List}} $
> > $ \small{\colorbox{orange}{def has$\underline{}$close$\underline{}$elements(numbers: List[float], threshold: float) -> bool:}} $
> > &nbsp;&nbsp;&nbsp;&nbsp;$ \small{\colorbox{LightCoral}{""" Check if in given list of numbers, are any} \colorbox{lightgreen}{two} \colorbox{LightCoral}{numbers closer to each other than given threshold.}} $
> > &nbsp;&nbsp;&nbsp;&nbsp;$ \small{\colorbox{orange}{>>> has$\underline{}$close$\underline{}$elements(}\colorbox{orange}{[} \colorbox{lightgreen}{1.0}\colorbox{orange}{,} \colorbox{lightgreen}{2.0}\colorbox{orange}{,} \colorbox{lightgreen}{3.0} \colorbox{orange}{],} \colorbox{lightgreen}{0.5} \colorbox{orange}{)}} $
> > &nbsp;&nbsp;&nbsp;&nbsp;$ \small{\colorbox{LightCoral}{False}} $
> > &nbsp;&nbsp;&nbsp;&nbsp;$ \small{\colorbox{orange}{>>> has$\underline{}$close$\underline{}$elements(} \colorbox{orange}{[} \colorbox{lightgreen}{1.0}\colorbox{orange}{,} \colorbox{lightgreen}{2.8}\colorbox{orange}{,} \colorbox{lightgreen}{3.0}\colorbox{orange}{,} \colorbox{lightgreen}{4.0}\colorbox{orange}{,} \colorbox{lightgreen}{5.0}\colorbox{orange}{,} \colorbox{lightgreen}{2.0} \colorbox{orange}{],} \colorbox{lightgreen}{0.3} \colorbox{orange}{)}} $
> > &nbsp;&nbsp;&nbsp;&nbsp;$ \small{\colorbox{LightCoral}{True}} $
> > &nbsp;&nbsp;&nbsp;&nbsp;$ \small{\colorbox{LightCoral}{"""}} $
> > &nbsp;&nbsp;&nbsp;&nbsp;$ \small{\colorbox{orange}{for i in range(len(numbers}}\small{\colorbox{LightCoral}{)):}} $
> > &nbsp;&nbsp;&nbsp;&nbsp;&nbsp;&nbsp;&nbsp;&nbsp;$ \small{\colorbox{orange}{for j in range(i +}}\small{\colorbox{lightgreen}{1}}\small{\colorbox{orange}{, len(numbers}}\small{\colorbox{LightCoral}{)):}} $
> > &nbsp;&nbsp;&nbsp;&nbsp;&nbsp;&nbsp;&nbsp;&nbsp;&nbsp;&nbsp;&nbsp;&nbsp;$ \small{\colorbox{orange}{if abs(numbers[i] - numbers[j]) < threshold:}} $
> > &nbsp;&nbsp;&nbsp;&nbsp;&nbsp;&nbsp;&nbsp;&nbsp;&nbsp;&nbsp;&nbsp;&nbsp;&nbsp;&nbsp;&nbsp;&nbsp;$ \small{\colorbox{orange}{return}} \colorbox{LightCoral}{True} $
> > &nbsp;&nbsp;&nbsp;&nbsp;$ \small{\colorbox{orange}{return}} \colorbox{LightCoral}{False} $
> >
> > As demonstrated, while the most relevant expert is primarily selected, different experts can be dynamically activated within an instance, as the routing operates at token-level, whereas self-specialization always leverages only a single expert for all tokens.
> >
> > We have updated the pdf. We greatly appreciate your valuable feedback and reconsideration. Thank you!

---

> ### Comment · Reviewer_eqDV · 2024-11-23
> **Further Question**
>
> Thanks for the example. Can I understand the synergy as follows: the synergy happens at the level of token-level dynamic routing.
> I think the authors have provided a good rebuttal. I will decide after I check the updated pdf.

---

> ### Author Response · Authors · 2024-11-23
> **Thank you for your question!**
>
> Yes, exactly! The synergy occurs at the token level through dynamic routing. We’re glad to hear that you found our rebuttal satisfactory. The visualized examples mentioned above have been included in the updated paper (Table 10 in the Appendix).
>
> Thank you for your valuable feedback and thoughtful consideration!

---

> > ### Comment · Reviewer_eqDV · 2024-11-23
> > **Raised Score**
> >
> > Thanks again. Raised soundness and overall assessment.

---

> > > ### Author Response · Authors · 2024-11-23
> > >
> > > Thank you for raising your score. Your insights and constructive feedback have been invaluable in refining our paper!

---

### Official Review · Reviewer_1Lag · 2024-11-03

**Soundness:** 3
**Presentation:** 3
**Contribution:** 2
**Rating:** 6
**Confidence:** 3

**Summary:**

Self-MoE constructs the corresponding experts modules to realize the self-specialization mechanism to improve the performances on non-specialized tasks and diverse benchmarks such as knowledge, reasoning, math, and coding.

**Strengths:**

1. Introducing the computation overhead realizes better performance improvement
2. Using the self-optimized routing activates the distinct domain-specific capabilities to help improve the performance shared base LLM
3. The presentation is good and the experiments are detailed

**Weaknesses:**

1. The comparison with related works such as  [1] is not enough. The concept of lora-moe has been introduced in this work.  This slightly affects the novelty of the work.

[1] Zadouri T, Üstün A, Ahmadian A, et al. Pushing mixture of experts to the limit: Extremely parameter efficient moe for instruction tuning[J]. arXiv preprint arXiv:2309.05444, 2023.

**Questions:**

1. How does the number of experts impact the downstream performance? If the knowledge for differenct experts is similar, is it possible to reduce the number of experts?
2. How does the model generate to the unseen tasks?

---

> ### Author Response · Authors · 2024-11-21
> **Response to Reviewer 1Lag**
>
> Thank you for your constructive feedback. We address each of your points below.
>
> ___
>
> > *The concept of lora-moe has been introduced in ..*
>
> While [1] does introduce a parameter-efficient MoE concept using LoRA, our approach with Self-MoE differs significantly. Self-MoE leverages self-generated data to construct specialized expert modules from scratch, without relying on extensive domain-specific human-labeled datasets or the existence of trained expert modules trained on such costly human-created data. Therefore, as discussed throughout the paper (Related Work, Appendix D.1., Table 5, and Table 7) direct comparisons against such models, which assume the labeled data or pre-trained modules, are not entirely feasible. Our method uniquely addresses the challenge of domain specialization without relying on the costly process of annotating large amounts of data, while resolving over-specialization and forgetting issues when combining experts.
>
> [1] Pushing Mixture of Experts to the Limit: Extremely Parameter Efficient MoE for Instruction Tuning
>
> ___
>
> > *How does the number of experts impact the downstream performance? If the knowledge for differenct experts is similar, is it possible to reduce the number of experts?*
>
> Great question. In Section 4.7 and Table 4, we provide an analysis of how the number of experts affects downstream performance. We observe that adding experts improves performance up to a certain point, after which the gains stabilize, as shown by the average performance increase from one to four experts below.
>
> | # of Experts | Performance |
> |---|------|
> | 1 | 43.6 |
> | 2 | 49.8 |
> | 3 | 52.9 |
> | 4 | 54.3 |
>
> When adding experts with distinct domain knowledge (e.g., reasoning, math, coding), the model benefits from targeted expertise, but redundant experts would not offer additional gains. In cases where knowledge overlap exists among experts, reducing the number of experts may indeed be possible without substantial performance loss.
>
> ___
>
> > *How does the model generate to the unseen tasks?*
>
> Regarding generalization, we conducted a test to evaluate Self-MoE’s performance on non-target tasks in Section 4.4 and Table 3, as summarized below:
>
> | Category                         | Benchmark     | Base LLM | Instance Merging | Self-MoE |
> |-----------------------------------|---------------|----------|-------------------|-------|
> | ***Target***                        |               |          |                   |       |
> | Academic Knowledge                | MMLU          | 58.4     | 62.6              | 65.6  |
> | Reasoning                         | BBH           | 56.1     | 57.6              | 61.1  |
> | Math                              | GSM8K         | 42.5     | 53.5              | 52.5  |
> | Coding                            | HumanEval     | 34.1     | 36.0              | 37.8  |
> |                | **Target Avg.**              | 47.8     | 52.4              | 54.3  |
> | ***Non-Target (In-Expertise)***     |               |          |                   |       |
> | Math                              | MATH          | 20.7     | 15.3              | 21.4  |
> | Coding                            | MBPP          | 37.8     | 37.6              | 39.6  |
> | ***Non-Target (Out-of-Expertise)*** |               |          |                   |       |
> | World Knowledge                   | Natural Questions | 24.2     | 22.3              | 24.5  |
> | TriviaQA                          | TriviaQA      | 63.9     | 58.6              | 62.5  |
> | Commonsense                        | Hellaswag     | 80.6     | 78.0              | 80.7  |
> | PIQA                              | PIQA          | 81.1     | 80.1              | 81.2  |
> | Safety                            | TruthfulQA    | 44.7     | 42.2              | 44.3  |
> |            | **Non-Target Avg.**               | 50.4     | 47.7              | 50.6  |
>
> The results suggest that Self-MoE not only improves performance on non-targeted but relevant tasks, but also preserves existing capabilities of the base model on out-of-expertise tasks, unlike the over-specialized models that exhibit significant drops outside their expertise. This is possible because Self-MoE’s diverse, self-specialized experts cover a range of capabilities, allowing the model to draw on relevant expertise even for unfamiliar tasks. By activating the most aligned expert based on task characteristics, Self-MoE retains robust performance across varied domains without over-specializing.

---

> ### Author Response · Authors · 2024-11-23
> **Looking Forward to Your Reply**
>
> Dear Reviewer 1Lag,
>
> As the discussion period is ending soon, we would greatly appreciate it if you could take a look at our response to your review and let us know if you have any remaining questions. Otherwise, we would really appreciate it if you could consider adjusting the score, given our response and update. We look forward to hearing from you and addressing any remaining concerns before the end of the discussion period.
>
> Best regards,
>
> Authors

---

### Official Review · Reviewer_tfML · 2024-11-05

**Soundness:** 3
**Presentation:** 3
**Contribution:** 3
**Rating:** 6
**Confidence:** 3

**Summary:**

This paper presents a novel approach to modularize large language models by constructing expert modules from self-generated synthetic data and creating a compositional system. Unlike previous Mixture of Experts approaches that use LoRA and rely on either human-labeled data or pre-trained modules, this method develops modules from scratch, adapting the model to specific tasks. Experimental results demonstrate the advantages of this approach over base LLMs and self-specialized LLMs in multiple tasks.

**Strengths:**

- Mitigation of Forgetting: Compared to monolithic models, which often face challenges with knowledge retention, the proposed approach maintains the integrity of each expert module, which enhances the overall model's performance and adaptability.

- Lightweight and Synthetic Data-Driven Modules: The method constructs individual, lightweight expert modules from synthetic data, which bypasses the need for human-labeled data and broadens the scope of applicability.

- Generalization: The generalization tests indicate that the Self-MoE approach offers benefits beyond the targeted tasks, improving performance on benchmarks that were not explicitly used in training.

**Weaknesses:**

- Limited Model Sizes: The experiments primarily focus on small-scale LLMs (7B/13B models), leaving open the question of whether these findings can extend to larger models. Testing on a wider range of model sizes would strengthen the claims.
- Quality and Diversity of Synthetic Data: How does the approach ensure the correctness, diversity, and quality of the instruction-response data generated for training the expert modules?
- Domain Granularity: How is the domain granularity determined? For instance, broad domains like reasoning, math, and coding can be subdivided further (e.g., reasoning into medical, finance). To what extent do the findings depend on the chosen level of domain specificity?

**Questions:**

- Router Details: The router is crucial to the Self-MoE model’s performance but is described only at a high level in the paper. More detailed information (e.g., training etc.) in the main body would enhance the clarity and accessibility of the approach.

- Synergies Across Domains: One reason for the Self-MoE’s superior performance might be that LLMs can exploit synergies across areas of expertise. Would this advantage persist if the domains were highly distinct, such as medical vs. finance?

---

> ### Author Response · Authors · 2024-11-21
> **Response to Reviewer tfML (Part 1)**
>
> Thank you for your thoughtful feedback and positive evaluation of our work.
>
> ___
>
> > *The experiments primarily focus on small-scale LLMs (7B/13B models)...*
>
> Thank you for this point. We agree that testing on larger-scale models would further strengthen our paper. Due to computational constraints, we focused on 7B/13B models for this study, while exploring different model families, sizes, and base performance levels to demonstrate the broad applicability of our approach. We plan to conduct experiments with larger-scale LLMs in the future to further investigate how our findings extend to these models.
>
> ___
>
> > *How does the approach ensure the correctness, diversity, and quality of the instruction-response data generated for training the expert modules?*
>
> In order to diversify the instructional data while ensuring being relevant and correct, we leverage a small set of human-labeled seed data as in-context examples. Specifically, the in-context seed demonstrations are randomly sampled (from the small seed set) and shuffled, which are then incorporated into the prompt, and instructions are generated through sampling with a high temperature. We believe that synthetic data diversification is an important, ongoing area of research that complements our Self-MoE approach but remains orthogonal to it. As methods for synthetic data generation advance, they can directly enhance Self-MoE’s performance with better self-specialized expert modules.
>
> To ensure the quality, we conducted extensive analyses of the self-synthesized datasets, focusing on diversity, complexity, and the model performance compared to human-labeled datasets. The findings are summarized below:
>
> First, we analyzed Type-to-Token Ratio (TTR), and semantic similarity of the pairwise instructions' embeddings using SBERT, as used in [1] for diversity measurement.
>
> | TTR (↑) | MMLU | BBH | GSM8K | HumanEval | Avg. |
> |------------------------------------|--------|-------|-------|--------|-------|
> | Human-Labeled Data | 0.2671 | 0.1672 | 0.1683 | 0.1121 | 0.1787 |
> | Synthetic Data           | 0.2639 | 0.1889 | 0.1484 | 0.0961 | 0.1743 |
>
> Synthetic data demonstrates comparable linguistic diversity to human-labeled data, with a slightly higher TTR for BBH, suggesting that the synthetic data includes richer lexical variation, especially in reasoning tasks. This indicates that the synthetic data contains a wider vocabulary or more diverse terms.
>
> | Semantic Similarity (↓) | MMLU | BBH | GSM8K | HumanEval | Avg. |
> |------------------------------------|--------|-------|-------|--------|-------|
> | Human-Labeled Data | 0.2625 | 0.1757 | 0.4125 | 0.4608 | 0.3279 |
> | Synthetic Data | 0.3129 | 0.1948 | 0.3360 | 0.4791 | 0.3307 |
>
> For semantic similarity, synthetic data achieves generally low similarity in absolute terms within each dataset, similar to human-labeled data (0.3307 vs. 0.3279). This suggests high semantic diversity overall, reflecting the natural diversity found in human-labeled data.
> Next, we leveraged a strong instruction-following model, GPT-4, as a judge to classify which instruction was synthetic. Given a pair of human-labeled and synthetic instructions, the classification accuracy was as follows:
>
> | Classification Accuracy (↓) | MMLU | BBH | GSM8K | HumanEval |
> |----------------------|--------|-------|-------|--------|
> | LLM as-a-judge | 55% | 68% | 60% | 50% |
>
> The classification accuracy ranged from 50% (random guessing) to 68%, with the lowest accuracy for HumanEval and MMLU, suggesting that synthetic data closely mimics human complexity in these domains. Conversely, the higher accuracy for BBH and GSM8K indicates that synthetic data in these domains has room to improve.
>
> Finally, we compared the performance of Self-MoE fine-tuned with synthetic data against human-labeled seed data. The results are summarized below:
>
> | Configuration | Knowledge | Reasoning | Math | Coding | Avg. |
> |------------------------------------|--------|-------|-------|--------|-------|
> | w/ Human-labeled data (Seed) | 57.4 | 57.0 | 45.0 | 34.1 | 48.4 |
> | w/ Synthetic data (1x)               | 57.7 | 55.9 | 45.5 | 32.9 | 48.0 |
> | w/ More Synthetic data (5x)      | 61.3 | 58.4 | 48.4 | 36.6 | 51.2 |
> | w/ More Synthetic data (50x)    | 65.6 | 61.1 | 52.5 | 37.8 | 54.3 |
>
> We observed that with the same quantity (400) as the seed, synthetic data achieves performance similar to human-labeled data in average. When scaling up the size (5x and 50x), synthetic data demonstrates the effectiveness and scalability of synthetic data in fine-tuning expert modules.
>
> [1] Improving Diversity of Commonsense Generation by Large Language Models via In-Context Learning

---

> ### Author Response · Authors · 2024-11-21
> **Response to Reviewer tfML (Part 2)**
>
> > *How is the domain granularity determined? …*
>
> Thank you for this insightful question. The current domains (e.g., reasoning, math, coding) were chosen to capture broad categories with distinct task demands, allowing each expert to focus on a coherent capability. We agree that more granular domains (e.g., medical reasoning, financial reasoning) could provide even finer specialization, potentially improving performance further. However, for the scope of this study, we opted for broader categories to balance performance and computational efficiency. We see domain granularity as a tunable parameter that could be explored in future work to better align expert modules with specific application needs.
>
> ___
>
> > *The router is crucial to the Self-MoE ... More detailed information (e.g., training etc.) in the main body would enhance the clarity and accessibility of the approach.*
>
> Thank you for the suggestion. We agree that the router is one of the central parts in our work, and additional details would clarify its role. The router is implemented as a linear layer shared across all LoRA layers. It is trained using the aggregated self-generated data to learn optimal expert selection based on task requirements. Importantly, the router is trained separately after the expert modules are trained and frozen, ensuring the explicit semantic distinction of the expert modules is preserved. We will expand its description in the main body to provide greater clarity and accessibility.
>
> ___
>
> > *Would Self-MoE’s advantage persist if the domains were highly distinct?*
>
> We appreciate this insightful question. While our current setup demonstrates synergies across broad domains, we hypothesize that Self-MoE’s modular design would still provide benefits even with highly distinct domains. In Section 4.4 and Table 3, we conducted a generalization test to evaluate Self-MoE’s performance on non-target tasks, as summarized below:
>
> | Category                         | Benchmark     | Base LLM | Instance Merging | Self-MoE |
> |-----------------------------------|---------------|----------|-------------------|-------|
> | ***Target***                        |               |          |                   |       |
> | Academic Knowledge                | MMLU          | 58.4     | 62.6              | 65.6  |
> | Reasoning                         | BBH           | 56.1     | 57.6              | 61.1  |
> | Math                              | GSM8K         | 42.5     | 53.5              | 52.5  |
> | Coding                            | HumanEval     | 34.1     | 36.0              | 37.8  |
> |                | **Target Avg.**              | 47.8     | 52.4              | 54.3  |
> | ***Non-Target (In-Expertise)***     |               |          |                   |       |
> | Math                              | MATH          | 20.7     | 15.3              | 21.4  |
> | Coding                            | MBPP          | 37.8     | 37.6              | 39.6  |
> | ***Non-Target (Out-of-Expertise)*** |               |          |                   |       |
> | World Knowledge                   | Natural Questions | 24.2     | 22.3              | 24.5  |
> | TriviaQA                          | TriviaQA      | 63.9     | 58.6              | 62.5  |
> | Commonsense                        | Hellaswag     | 80.6     | 78.0              | 80.7  |
> | PIQA                              | PIQA          | 81.1     | 80.1              | 81.2  |
> | Safety                            | TruthfulQA    | 44.7     | 42.2              | 44.3  |
> |            | **Non-Target Avg.**               | 50.4     | 47.7              | 50.6  |
>
> The results suggest that Self-MoE not only improves performance on non-targeted but relevant tasks, but also preserves existing capabilities of the base model on out-of-expertise tasks, unlike the over-specialized models that exhibit significant drops outside their expertise. By activating the most aligned expert for each input token based on task characteristics, Self-MoE retains robust performance across varied domains without over-specializing, which we believe would extend to highly distinct domains, avoiding a forgetting issue seen in monolithic models.

---

### Author Response · Authors · 2024-11-22
**General Response to Reviewers**

We thank all the reviewers for taking their valuable time to provide insightful comments and suggestions to improve our paper.

We are pleased to note that all reviewers recognized the value of our work, as evidenced by the following positive comments:
- “It's a good idea to stimulate diverse domain-specific competencies inherent within the base model …” (fc1R)
- “Novel idea of self-MOE and it can be a promising new paradigm for stronger foundation models.” (eqDV)
- “This approach offers the advantage of eliminating the necessity for domain-specific data acquisition.” (fc1R)
- “Introducing the computation overhead realizes better performance improvement” (1Lag)
- “Using the self-optimized routing activates the distinct domain-specific capabilities to help improve the performance shared base LLM” (1Lag)
- “Mitigation of Forgetting”, “Lightweight and Synthetic Data-Driven Modules”, “Generalization” (tfML)
- “I actually like this paper.” (eqDV)
- “The writing and presentation are clear and easy to read.” (fc1R)
- “Paper is well-written.” (eqDV)
- “The presentation is good and the experiments are detailed” (1Lag)

We believe the thoughtful reviews and the recommendations made by the reviewers have substantially improved the quality of the paper. Based on the reviewers' suggestions, we have updated the paper with the following major changes:
- Added a new experiment on joint training with top-2 activation (Table 2 and Section 4.2) (eqDV)
- Included a new ablation result on layer-specific routers, comparing with a shared router (Table 2 and Section 4.2) (fc1R)
- Incorporated Instance Merging results for different base models (Figure 4 and Section 4.5) (fc1R)
- Added new analyses of self-generated data (Table 5 and Section 4.9) (fc1R)
- Reorganized Figure 1 and deleted Table 5 (full content in Table 7) to secure space for additional results (eqDV)
- Elaborated on the router in Section 3.2 (tfML)
- Elaborated on the baseline comparisons in Section 4 (fc1R)
- Provided detailed results of Self-MoE applied to different base LLMs (Table 8 in Appendix) (eqDV)
- Included visualized examples of the token-level routing decisions (Table 10 in the Appendix) (eqDV)

We hope that these changes adequately address your comments and improve the clarity and the quality of the paper. Thank you again for your thoughtful feedback and continued consideration of our work!

---

### Meta-Review · Area_Chair_TYJq · 2024-12-19

**Metareview:**

This paper proposes a novel approach to transform monolithic LLMs into a mixture-of-experts system with self-specialized experts using synthetic data. The key scientific claim is that Self-MoE can improve model performance across diverse domains while mitigating catastrophic forgetting, without requiring domain-specific human-labeled data. The main strengths are: (1) the novel combination of self-specialization with MoE architecture, (2) effective use of synthetic data for expert training, and (3) comprehensive empirical validation across multiple base models and tasks. The primary weaknesses include limited experimentation with larger models and some initial ambiguity around the router training mechanism. I recommend acceptance due to its novel technical contribution in combining self-specialization with MoE, strong empirical results showing consistent improvements across different base models, and the practical advantage of not requiring domain-specific labeled data.

**Additional Comments On Reviewer Discussion:**

During the discussion phase, reviewers raised several key points: (1) router training details and expert selection mechanism, (2) comparison with related LoRA-MoE works, (3) quality and diversity of synthetic data, and (4) ablation studies on different routing configurations. The authors addressed these concerns by providing analyses of synthetic data quality, additional experiments on joint training and layer-specific routers, and clarification of token-level routing mechanisms. After the discussion, three reviewers raised their scores based on the authors' responses and paper updates, while one maintained their score. The clarification of token-level routing and synthetic data analysis were particularly convincing in demonstrating the method's effectiveness. There are still remaining concerns about model scale limitation, synthetic data quality and domain granularity, but these limitations primarily point to future research direction rather than fundamental flaows. The thorough empirical validation and responses to main technical concerns support the decision to accept this submission.

---

### Decision · Program_Chairs · 2025-01-22

Accept (Poster)